**Data Availability Statement:** All relevant data are within the paper and its Supporting Information files.

**Funding:** This work was carried out with the support of "Cooperative Research Program for Agriculture Science & Technology Development

# UV-LED lights enhance the establishment and biological control efficacy of *Nesidiocoris tenuis* (Reuter) (Hemiptera: Miridae)

Young-gyun Park[1]¤, Joon-Ho Lee[1,2]*

**1** Department of Agricultural Biotechnology, Entomology Program, Seoul National University, Seoul, Republic of Korea, **2** Research Institute of Agriculture and Life Sciences, Seoul National University, Seoul, Republic of Korea

¤ Current address: Department of Plant Medicals, Andong National University, Andong, Republic of Korea
* jh7lee@snu.ac.kr

## Abstract

The zoophytophagous mirid *Nesidiocoris tenuis* (Hemiptera: Miridae) is one of the biological control agents against the whitefly *Bemisia tabaci* (Hemiptera: Aleyrodidae), a major pest of greenhouse crops. The successful establishment of a biological control agent and its co-occurrence with the target pests increases the efficacy of biological control programs in greenhouses. In this study, we explored the effects of different wavelengths of LED light on establishment of *N. tenuis* in laboratory condition, with the goal of enhancing the biological control of *B. tabaci* in greenhouse crops. *Nesidiocoris tenuis* was most strongly attracted by LED light at a wavelength of 385 nm. This same wavelength was also highly attractive to *B. tabaci* in Y-tube experiments with lights of specific wavelengths provided is each arm of the apparatus. In trials in growth chambers, we verified the attraction of *N. tenuis* to 385 nm wavelength. When LED light at a wavelength of 385 nm was used in a growth chamber for 6 hours out of 24 hours, it significantly increased the remaining number of *N. tenuis* in growth chamber and level of predation compared to treatment with white LED light or without LED light. In conclusion, UV-LED light at a wavelength of 385 nm attracts both *B. tabaci* and *N. tenuis*. Thus, it would be used for enhancing early establishment of this mirid bug, better spatial congruence of both mirid bug and whitefly, and better control of the whitefly.

## Introduction

*Bemisia tabaci* (Gennadius) (Hemiptera: Aleyrodidae) causes significant damage to horticultural crops such as tomato, sweet pepper, and flowers, etc. [1–3], both by directly feeding on crop plants and by producing honeydew, which leads to sooty molds that reduce photosynthesis [4, 5]. Even more importantly, this whitefly vectors plant viruses such as the tomato yellow leaf curl virus, which causes serious economic loss in tomato [6]. Chemical insecticides have been commonly used to control *B. tabaci*. Despite their popularity, chemical insecticides are becoming increasingly less effective due to the development of resistance by this insect pest [7, 8]. Therefore, alternative control strategies such as combinations of pesticides and plant

(Project No. PJ01258303)", Rural Development Administration, Republic of Korea, "Cooperative Research Program for Agriculture Science & Technology Development (Project No. 319007-01-1-HD060)", Institute of Planning and Evaluation for Technology in Food, Agriculture, and Forestry, Republic of Korea, and "Brain Korea 21 Plus", Ministry of Education, Republic of Korea.

**Competing interests:** The authors have declared that no competing interests exist.

extracts, manipulations of tri-trophic effects, and biological control have become of increasing interest for control of *B. tabaci* [9–14].

*Nesidiocoris tenuis* (Reuter) (Hemiptera: Miridae) is a generalist predator that has been used for pest biological control of mites, thrips, whiteflies, aphids, and some moths in both greenhouses and outdoor crops [15–19]. In greenhouses, this mirid effectively controls the major insect pests of tomato, including whiteflies and tomato borers [20–28]. In one day at 25°C, *N. tenuis* can eat >30 pupae of *B. tabaci* or >50 eggs of *Tuta absoluta* (Meyrick) (Lepidoptera: Gelechiidae) [16, 29]. However, due to its zoophytophagous nature of its feeding activity, *N. tenuis* also attacks plants during periods of low prey density [30, 31]. Damage to crops, however, is often slight in comparison to the gain from whitefly suppression, supporting the continued use of *N. tenuis* as a biological control agent [24]. Related studies have been conducted to reduce the mirid's risk to crop plants, including studies of plant resistance to *N. tenuis* mediated by endophytic strain of *Fusarium solani* K [32], application of sugar as a complementary alternative food, and optimizing the release density for *N. tenuis* [30, 33]. Because generalist natural enemies can subsist on alternative food sources when the target pest is rare, they can be more adaptable to changing circumstances in the crop than specialist natural enemies [34].

Successful establishment and persistence of natural enemy populations in areas where pests are most concentrated must be achieved for biological control agents to be effective in greenhouses. These outcomes can be promoted by having proper environmental conditions and prey densities as needed by the biocontrol agent [24, 35, 36]. Alternative food sources and banker plants can be used to help conserve natural enemies [37, 38]. Also, it is helpful to minimize emigration of natural enemies leaving the crop and to maximize natural enemy aggregation where pests are most abundant. Manipulations tested to improve the spatial association of natural enemies with the pests they attack include the release of the natural enemies onto the transplant trays (when plants are most concentrated) a few days before planting and the application of supplementary foods such as plant nectar or sugar to areas with high pest infestations [21, 22, 33, 35, 39, 40].

A further tool with potential to manipulate the spatial distribution of natural enemies in greenhouses is the use of specialized light sources. Insects can be attracted or repelled by light, depending on the wavelength [41–43]. The positive phototaxis behavior of some natural enemies has been explored as a tool to improve biological control using lights, mainly in Japan [15, 44, 45]. For example, thrips can be successfully controlled using violet light-emitting diode (LED) lights, which improve the establishment rate of *Orius sauteri* (Poppius) (Hemiptera: Anthocoridae) in eggplant crops outdoors [44]. Given that this strategy [44] was successful in open agricultural fields, we hypothesized that the same use of LED light sources should work equally well in greenhouses. Our goal, therefore, was to evaluate the use of LED lights for enhancement of the impact of *N. tenuis* in the laboratory condition for pest (i.e., *B. tabaci*) control. However, the reaction of key pest species to selected wavelength also have to be considered. If a selected LED light is asymmetrically attractive between *N. tenuis* and its prey, the spatial separation between the predator and the prey, rather than being reduced, might increase. Therefore, we hypothesized that the best strategy would be to select LED light wavelengths that have high attractiveness for both the predator and prey. The specific objectives of this study, therefore, were to determine if selected wavelengths of LED light would enhance establishment of *N. tenuis*. Additionally, the responses of *B. tabaci* to selected wavelengths of LED light were explored to determine their impact on the spatial congruence of the distributions of natural enemies and the target pest.

## Materials and methods

### Insects sources and rearing

For wavelength selection experiment (experiment 1), *N. tenuis* adults (300 individuals per one bottle) were purchased from Osang Kinsect System (Guri, Korea). After delivery, they were stored in an incubator (500 x 500 x 500 mm$^3$; width x length x height; Hanbaek Scientific Co., Suwon, Korea) at 25˚C for about 1 h before the experiments. For use in the incubator experiments (experiments 2 and 3), *N. tenuis* were reared for about four months in an acrylic cage (300 x 300 x 300 mm$^3$; width x length x height) in an incubator at 27.1 ± 0.22˚C, 68.9 ± 3.76% relative humidity (RH), and 14:10 (L:D) h photoperiod. We placed five or six tomato plants (about 30 cm in height) in the rearing cage for mirid oviposition. For mirid food, we supplied eggs of *Cadra cautella* (Walker) (Lepidoptera: Pyralidae). Tomato plants were replaced monthly, and new *C. cautella* eggs were provided twice each week. Eggs of *C. cautella* were purchased from Osang Kinsect System and kept in a freezer until needed. *Bemisia tabaci* adults were obtained from Gyeonggi-do Agricultural Research and Extension Service in Hwaseong, Korea, and reared in cages (300 x 300 x 300 mm$^3$; width x length x height) in the insectary at 24–25˚C, 60–70% RH, and 16:8 (L:D) h photoperiod. Tobacco plants (about 30 cm in height) were supplied to whiteflies for their feeding and oviposition. These plants were replaced at intervals of one or two months.

### Experiment 1: Wavelength selection (in Y-tube tests)

Wavelength selection experiments were conducted to find the wavelength of LED light that would attract both *N. tenuis* and *B. tabaci* by using a Y-tube assay (200 mm in length and 40 mm in diameter for each arm) [46]. Insects were introduced in the stem of the Y-tube, and the two arms transmitted light of specific wavelengths. At the end of each arm, a transparent sticky trap (40 mm diameter) was installed. At the center of each of these transparent sticky traps, a hole (7 x 7 mm$^2$; width x length) was drilled to reduce the interference of the trap for the light single. The LED lights were installed in cups that inserted tightly into the Y-tube arms' ends (45 mm external diameter). The distance between the transparent sticky traps and the LED light source was 10 mm. One light source emitted a white LED light (color temperature 5000K) as the control. In the other Y-tube arm, there was a test wavelength LED light. All LED lights were made by LG Innotek Co. Ltd. in Seoul, Korea. All LED equipment used in this test were made by Skycares in Gimpo, Korea. During a test, each arm of the Y-tube was capped to prevent penetration of light from the outside into the test arena. The body of the Y-tube was made of plastic (3D printing) that did not permit light to pass. Each wavelength assay was replicated five times. For each replication, insects were given one hour under at 25˚C to respond to light signals. For tests with *N. tenuis*, we examined 10 wavelengths (365, 385, 395, 405, 415, 425, 445, 495, 525, and 590 nm). For each replication, about 30 *N. tenuis* were randomly selected from the purchased bottle and placed in the central chamber. Five wavelengths (365, 385, 395, 405, and 445 nm) of LED light that attracted *N. tenuis* were also tested for *B. tabaci*. For each replication, approximately 50 *B. tabaci* adults were randomly selected from the rearing colony and placed in the central chamber. After the allotted test time of 1 hr elapsed, insects stuck on the transparent sticky trap and the number remaining in the central chamber of the Y-tube were counted. These numbers were divided by the initial number of insects released to calculate the rates of attraction for the treatment or control (on traps) and the rates of insects that failed to move out of the central chamber (non-responders).

## Experiment 2: No-choice test of effect of light treatment on *N. tenuis* movement and predator efficiency

To verify the attractiveness of 385 nm wavelength LED light (the optimal wavelength in the Y-tube assay), this light was compared to white LED (5000K) light or non-LED in an incubator assay. The treatment being assessed (385 nm) is consider within the group of wavelengths called UV-A, which are known to have no harmful effects on crops [47]. Thus, we did not have to considered the risk of other wavelength groups of UV light. In this experiment, we measured the rate of predation by the released mirids on eggs of *C. cautella* as a "predation index". The incubator was not a fully enclosed space. Rather, it had some holes in the inner side wall for environmental controls, and there was a vessel of water behind one wall. Some *N. tenuis* left the experimental arena and drowned in this water. Thus, we also counted the remaining number of mirids in an incubation chamber at the end of each assay. The experiment was run at $27.0 \pm 0.16°C$ and $68.8 \pm 1.82\%$ RH in an incubator (500 x 500 x 500 $mm^3$; width x length x height). In the incubator, a 30-W circular fluorescent light was installed on the ceiling and set for a 14:10 (L:D) h photoperiod.

Each run lasted 24 hours. In experiment 2, there were three treatments: 385 nm wavelength LED light, white LED (5000K) light, and a non-LED, which used only fluorescent light on the ceiling. There were ten replicates of each treatment. In each run, an acrylic cage (130 x 120 x 200 $mm^3$; width x length x height), was placed at each corner of the incubator. The two exposed sides of each cage had an unscreened rectangular hole (100 x 165 $mm^2$; width x length) to allow access for movement of *N. tenuis* among cages. In each cage, we placed a large Petri dish (100 mm x 42 mm; diameter x height; SPL life sciences, Pocheon, Korea) with water-saturated cotton. In each Petri dish, a tomato stem (about 90 mm in length with three leaves) was laid on the cotton and a tomato stem (about 110 mm in length with five leaves) was inserted into the cotton to provide refuge and food substrate for *N. tenuis*. To measure predation, feeding on a non-pest was used as a predation index to compare treatments. This index was the rate of predation on 100 eggs of *C. cautella*, which were placed on a small Petri dish (40 mm x 6 mm; diameter x height), which itself was placed on the large Petri dish (100 mm x 42 mm; diameter x height) on which the tomato stems had been placed.

In each run of the test, there was one light tested (i.e., either the 385 nm or the white LED (5000K) lights) and there were three control cages with just non-LED lighting. In each run, one of the four acrylic cages (the one with the light being tested) had a hole cut in the roof and a 3-W LED light appropriate for the treatment was set in the hole. The LED light was illuminated for a total of six hours (one hour of LED light together with fluorescent light and five hours of LED light only). The timing of when the additional lighting occurrence was based on data from a preliminary test in a greenhouse (Fig 1). The location of the cage receiving the LED lighting (i.e., the treatment) was moved to another corner of the incubator with each new replication. The three control cages were rotated clockwise as the LED lighting cage changed location between replicates. The control cages received only fluorescent light from the ceiling light with a photoperiod of 14:10 (L:D) h.

In each replicate of the test of each treatment, ten randomly selected *N. tenuis* from the colony were collected into a Petri dish (100 mm x 42 mm; diameter x height) that had water-saturated cotton and a tomato stem (about 90 mm in length) with three leaves without food (*C. cautella* eggs). The Petri dish was then placed in the center of incubator, and the *N. tenuis* bugs were released.

After 24 hours, the numbers of uneaten eggs of *C. cautella* and live *N. tenuis* adults found in each cage in the incubator were counted. Also, we recorded the number of *N. tenuis* that remained in the Petri dish in the center of the incubator where they had originally been released. These measured defined, respectively, the number of *N. tenuis* that moved around

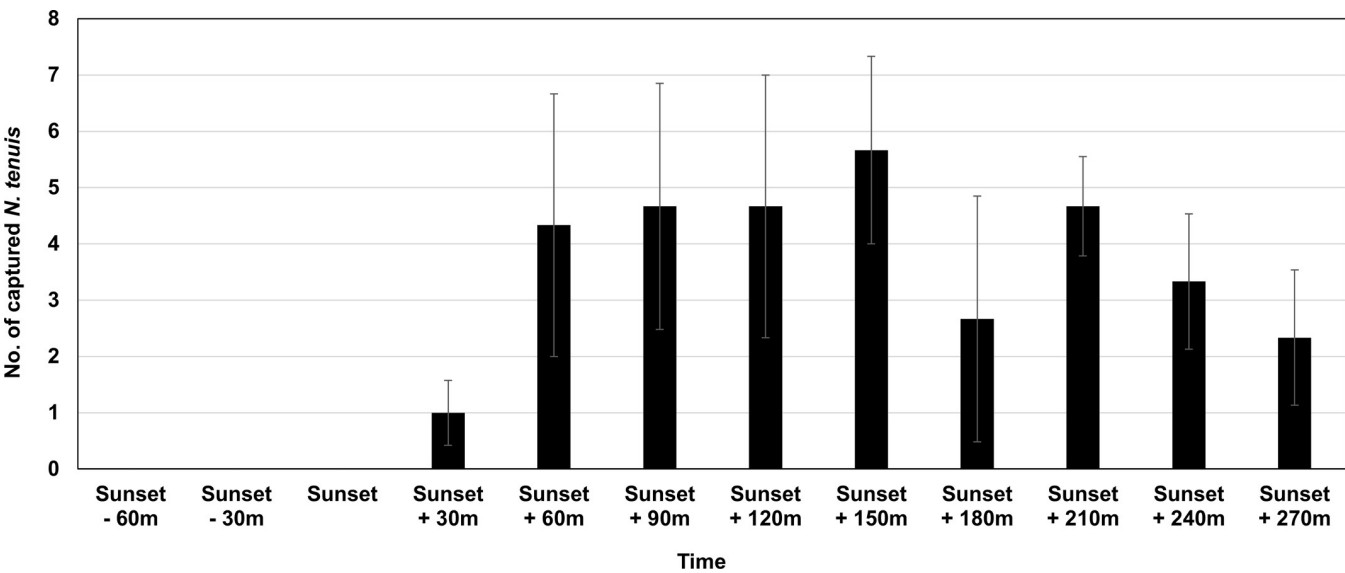

**Fig 1. Number (Mean ± S.E.) of *N. tenuis* adults attracted to 385 nm wavelength LED light (9-W) developed in a greenhouse (14.5 x 7 m². width x length) without plants (three replications, 100 *N. tenuis* in each replication) at different points in the daily light cycle.**

and participated in the trial and did not leave the incubator, and, for those still in the release dishes, the number of bugs that were "non-responders" that did not participate in the trial.

## Experiment 3: Choice test of effect of light treatment on *N. tenuis* movement and predator efficiency

Experiment 3 differed from experiment 2 in that the treatments were run together using a multiple choice design. The same two LED treatments were examined as in experiment 2 (385 nm and white light at 5000K). As in experiment 2, there was one control, being non-LED fluorescent lighting. Unlike experiment 2, there were three, not four, cages in the chamber at the same time: one 385 nm wavelength LED light, one white LED (5000K) light, and one non-LED. The experiment was repeated ten times, and 15 *N. tenuis* were used for each replication. Other experimental conditions were the same as stated above.

## Data analysis

In experiment 1 (wavelength selection), data were arcsine transformed and analyzed by analysis of variance (ANOVA) using PROC ANOVA in SAS [48]. Mean separation was conducted using Tukey's studentized range test.

In experiments 2 and 3 (Effects of LED light under no-choice and choice conditions), the percentages of predation of eggs of *C. cautella* and the number of *N. tenuis* remaining in incubator were arcsine transformed and analyzed by ANOVA using PROC ANOVA in SAS [48]. Mean separation was conducted using Tukey's studentized range test.

## Results

### Experiment 1: Wavelength selection (in Y-tube tests)

There was a significant effect of wavelength on the attraction of *N. tenuis* (Test LED, $F_{9, 40}$ = 31.02, $P < 0.001$; Non-response, $F_{9, 40}$ = 7.82, $P < 0.001$; Control LED, $F_{9, 40}$ = 24.11, $P < 0.001$; Fig 2). The attraction rate of *N. tenuis* was the highest at 385 nm wavelength (73.5%), followed

by 365 nm wavelength (62.6%). Attraction rates of *N. tenuis* to control LED light were >50% at wavelengths of 495, 525, and 595 nm. The five wavelengths that attracted *N. tenuis* most were the following, in order of attraction: 385 nm > 365 nm > 445 nm > 395 nm > 405 nm. Thus, these bug-attractive wavelengths were selected and tested against the pest, *B. tabaci*. The attraction rate of the whitefly to the tested LED light wavelengths decreased as the wavelength increased from 365 to 445 (Fig 3). No significant difference in attraction rate was found among wavelengths except for 445 nm, which showed the lowest attraction rate (37.6%) (Test LED, $F_{4, 20}$ = 4.40, $P$ = 0.010; Non-response, $F_{4, 20}$ = 2.27, $P$ = 0.098; Control LED, $F_{4, 20}$ = 3.49, $P$ = 0.026). Thus, the 385 nm wavelength was selected for the incubator assays.

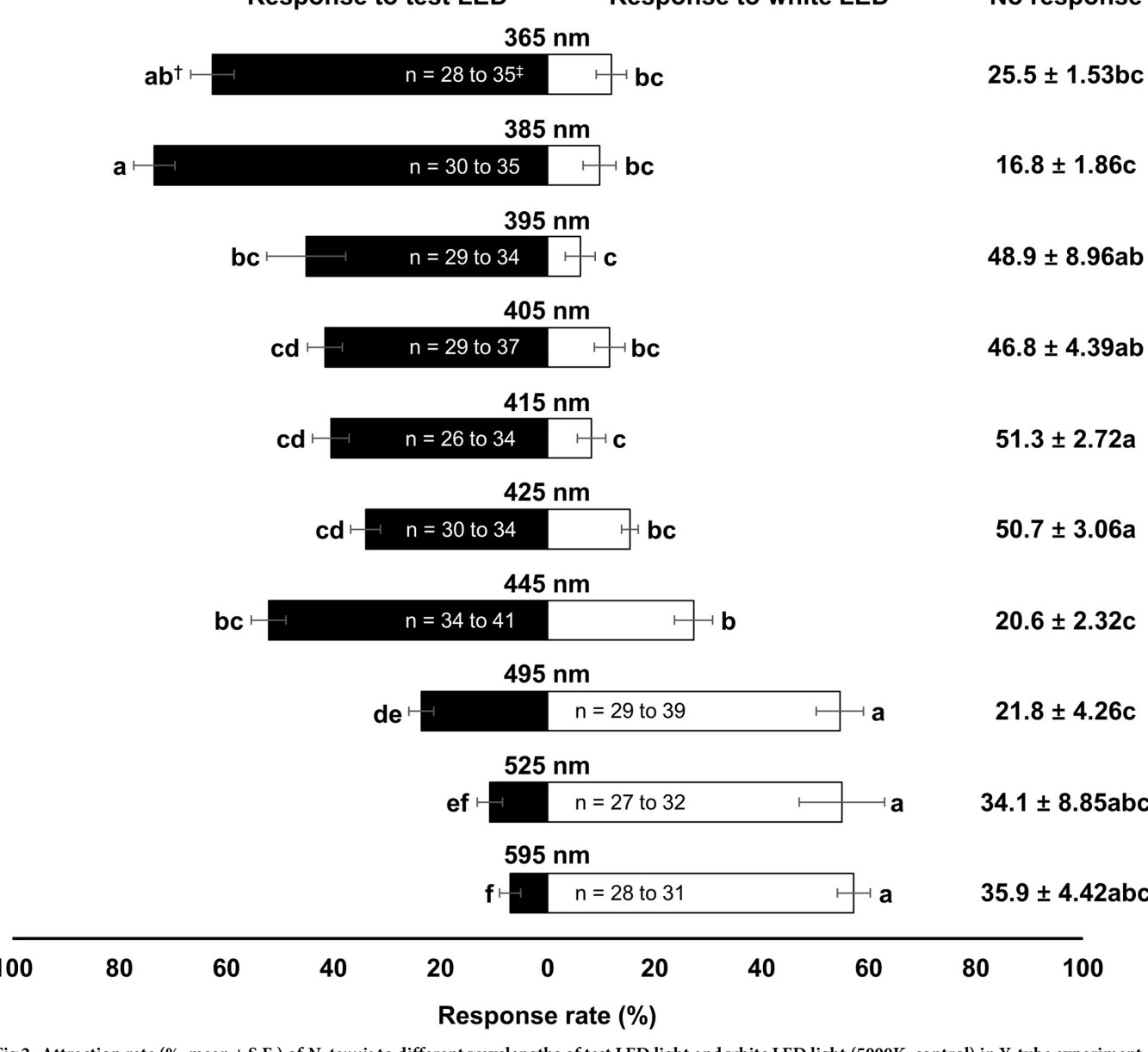

**Fig 2. Attraction rate (%, mean ± S.E.) of *N. tenuis* to different wavelengths of test LED light and white LED light (5000K, control) in Y-tube experiment (experiment 1).** [†]Means followed by the same letter within each response rate are not significantly different at α = 0.05, Tukey's studentized range test. [‡]Initial number range of insect that used in each replication of each treatment.

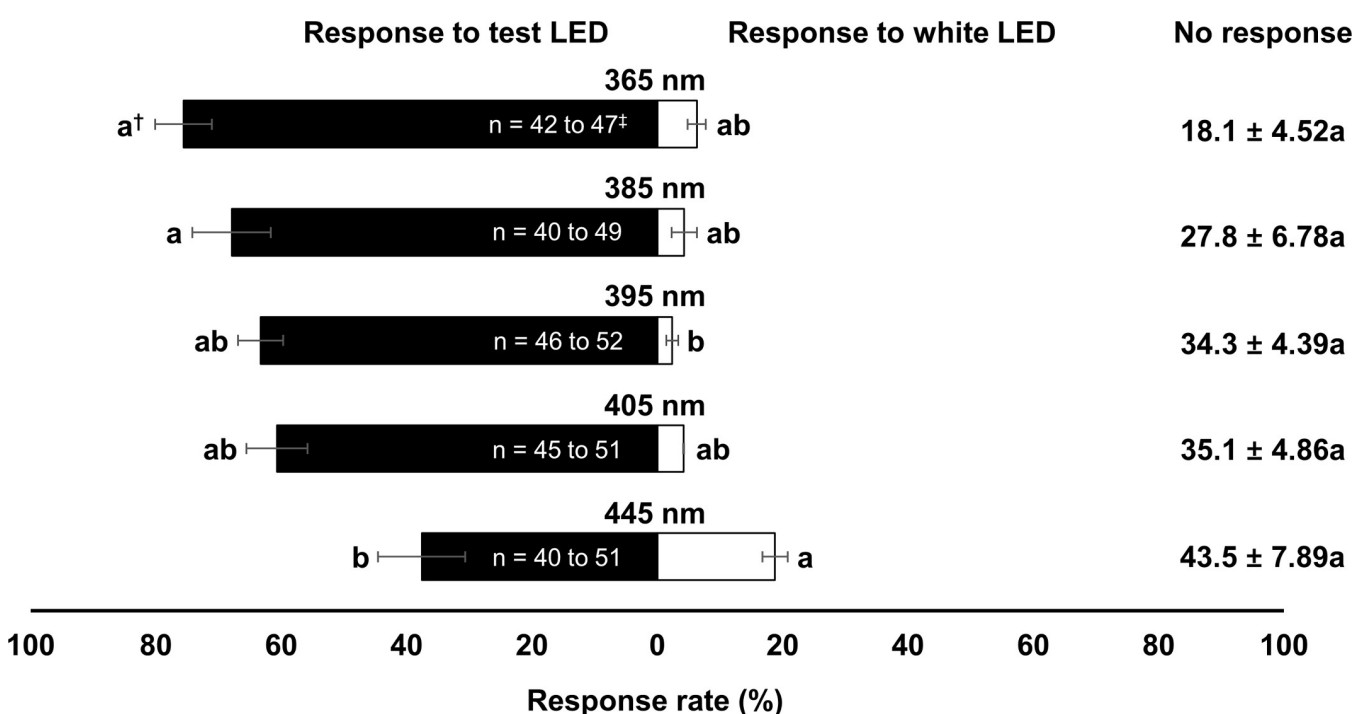

**Fig 3. Attraction rate (%, mean ± S.E.) of *B. tabaci* to different wavelengths of LED light and white LED light (5000K, control) in Y-tube experiment (experiment 1).** [†]Means followed by the same letter within each response rate are not significantly different at α = 0.05, Tukey's studentized range test. [‡]Initial number range of insect that used in each replication of each treatment.

## Experiments 2 and 3: No-choice and choice tests of effect of light treatment on *N. tenuis* movement and predator efficiency

When treatments were presented separately (experiment 2), LED light at 385 nm wavelength significantly affected the rate of predation of bugs on eggs of *C. cautella* and the number of *N. tenuis* that remained in the release area. Numbers of eggs consumed under conditions of 385 nm LED, white LED (5000K) light, and the non-LED treatment were 121.5, 52.4, and 67.0 eggs, respectively (Fig 4) ($F_{2, 27}$ = 8.17, $P$ = 0.002). Total numbers of *N. tenuis* remaining in the incubator in 385 nm, white LED (5000K), and non-LED treatment groups were 5.9, 3.7, and 3.5, respectively ($F_{2, 27}$ = 9.64, $P$ < 0.001). *Nesidiocoris tenuis* remained in the release area longer in the 385 nm treatment group than in other treatments.

When the same treatments were presented under choice conditions (experiment 3), predation levels and the number of *N. tenuis* remaining were both significantly higher in the 385 nm treatment cage (Table 1) (Predation level, $F_{2, 27}$ = 3.90, $P$ = 0.033; Number of *N. tenuis* remaining, $F_{3, 36}$ = 3.61, $P$ = 0.022).

## Discussion

The studies for enhancing the impact of *N. tenuis* have investigated various supplements and manipulations including the use of plant volatiles, pre-planting releases, and provision of supplementary food [21, 22, 33, 49]. Here, we report evidence of potential value of UV-LED to enhance this mirid bug. Many of the key pests of greenhouses–thrips, whiteflies, and aphids [50–56]–are attracted to UV light [42, 50, 51, 57], likely because UV light often plays an important role in insect movement such as navigation and orientation [50, 51]. The use of UV light for pest suppression has been explored in greenhouse crops [43, 58]; light traps can attract and

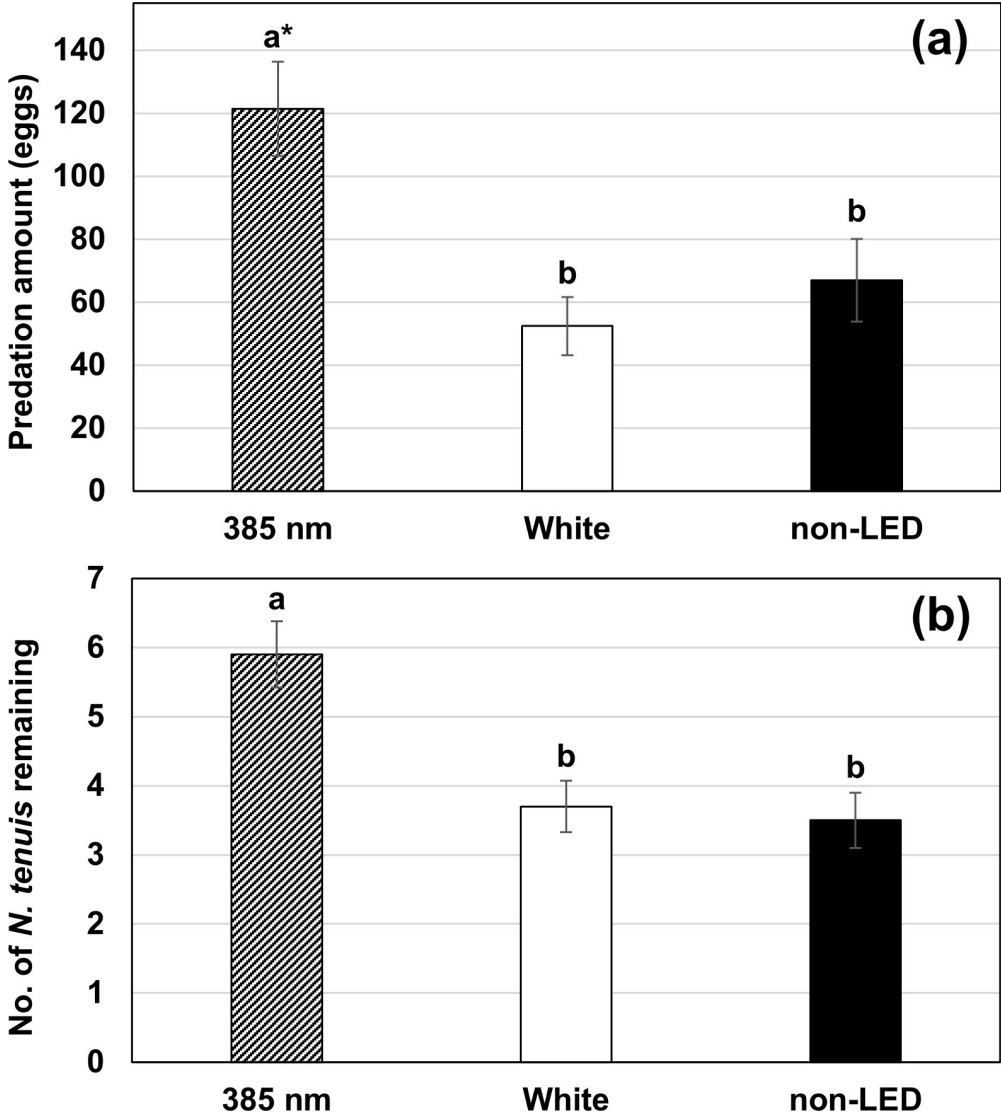

**Fig 4.** Level of predation on eggs of *C. cautella* (a) and number of *N. tenuis* bugs remaining in the release area (b) for 385 nm wavelength LED light, white LED light (5000K), and non-LED treatment groups (mean ± S.E.) (experiment 2). *Means followed by the same letter in graphs were not significantly different at α = 0.05, in Tukey's studentized range test.

**Table 1. Predation amount and number of *N. tenuis* remaining in the condition of same time treatment of 385 nm wavelength LED light, white LED light (5000K), and non-LED (mean ± S.E.) (experiment 3).**

| Treatment cage | Predation amount | *N. tenuis* remaining |
|---|---|---|
| 385 nm LED | 41.6 ± 10.84a* | 2.9 ± 0.53a |
| White | 22.5 ± 7.42ab | 2.2 ± 0.53ab |
| non-LED | 11.7 ± 3.44b | 0.8 ± 0.25b |
| Center | - | 1.6 ± 0.37ab |

*Means followed by the same letter in each column are not significantly different at α = 0.05, Tukey's studentized range test.

capture pests directly and UV absorbing films can interrupt the movement of pest into greenhouses [43, 58–60]. Furthermore, UV-LED light can be added to sticky traps to enhance their attractiveness for *Frankliniella occidentalis* (Pergande) (Thysanoptera: Thripidae) [59], and near UV-LED light can enhance the establishment of natural enemies [15, 44, 45].

In this study, *B. tabaci* and *N. tenuis* showed similar preferences among wavelengths. *Nesidiocoris tenuis* was most attracted in our study to 385 nm wavelength, while *B. tabaci* preferred 365 nm wavelength the most followed by its attraction to 385 nm wavelength without showing statistically significant (Figs 2 and 3). Therefore, we chose 385 nm wavelength for further tests and accomplishing their spatial congruence. However, Uehara et al. [15] reported preferences of *N. tenuis* in the 385 to 405 nm wavelengths, with mated males favoring the higher end of this range. This difference been studies might be due to different light quality, insect strain, and test method or the fact that Uehara et al. [15] used a multiple choice design, while we use a single choice test against control design.

In greenhouses, blue (near 450 nm) and red (near 650 nm) lights are often used as supplementary light to promote plant growth [61–64]. However, we found that for *N. tenuis* 445 nm wavelength was less preferred than 385 nm wavelength, which might reduce this bug's impact for biological control (Fig 2). In addition, we found that the attraction of *N. tenuis* to light drastically decreased at longer wavelengths, suggesting that red light (near 650 nm) actually suppress or interfere with the use of this bug for biological control. In our study, longer wavelengths (495, 525, and 595 nm) in a Y-tube experiment were less attractive than white LED light (5000K). These reactions might be due to avoidance behavior of the mirids. Lights of 505 to 530 nm wavelengths are used for growth and photosynthesis in tomato and cucumber, and therefore use of *N. tenuis* may be unsuccessful under those conditions [64]. Supplementary light for promoting plant growth often based on a combination of different wavelengths of light [61, 63]. Thus, when using such supplementary light, an interference effect from red light on both insect pests and natural enemies might occur. Further study is needed to elucidate potential effects of different wavelengths used as supplementary light for plant growth on insect pests and natural enemies.

We verified 385 nm wavelength LED light to be attractive to *N. tenuis* in laboratory incubator assays. Total predation (measured as predation on egg of a stored product moth) was twice high in the 385 nm wavelength LED light group compared to the non-LED or control LED light group (Fig 4). Also, the number of *N. tenuis* the remained in the release area was greatest in the 385 nm wavelength LED light group (Fig 4), and it was likely this longer retention time of *N. tenuis* at 385 nm wavelength that led to increased predation. Thus, if this LED light is applied in the greenhouse, it may increase the retention time of *N. tenuis* resulting in enhanced establishment of this predator. Although the periodicity of special lighting was set to minimize the interference from other light (see Fig 1), the effect of 385 nm wavelength was reduced when it was used with other light that had some attractive effect (Table 1). Thus, growers should avoid the use of wavelengths other than UV at the same time.

Ogino et al. [44] have reported that the establishment of *O. sauteri* released for thrips control was enhanced when 405 nm wavelength LED light was applied in eggplant fields. Although supplementing the lighting in a greenhouse (or field) with a short wavelength light may be useful for biological control, some caution is needed because these lights also may attract the pests [42]. Thus, the greenhouse's vents should be closed during the supplemental lighting period to reduce the attraction of whiteflies into the greenhouse, and regular monitoring of both natural enemy and pest densities would be needed. Further studies are needed to elaborate methods of applying short wavelength light in fields.

Results of this study clearly indicate that 385 nm wavelength LED light can improve the efficacy of *N. tenuis* in released area. Also, the 385 nm wavelength LED light could help achieve

the spatial congruence of both *N. tenuis* and *B. tabaci* in the target area because this LED light can attract both species. For successful biological control in greenhouses, proper establishment of a released biological control agent and its spatial aggregation to areas with the insect pest are very important. In this regard, using attractive LED light at an effective wavelength (such as 385 nm in our study) could be useful for enhancing the effectiveness of biological control agents. The use of LED light may also be more effective when is combined with other control tactics such as banker plant for natural enemy habitat, plant attractive volatiles, alternative foods, and method of mediating plant resistance to *N. tenuis* to reduce feeding damage [24, 32, 33, 37, 38, 49]. Further studies are needed to better understand how best to use LED lights, such as optimal lighting time and wattage of lights and assessments of the applicability of such light supplementation in the greenhouse.

## Supporting information

**S1 Data.**
(XLSX)

## Acknowledgments

We are grateful to Dr. U. T. Lim (Andong National University) for improving the quality of our manuscript.

## Author Contributions

**Conceptualization:** Young-gyun Park, Joon-Ho Lee.

**Data curation:** Young-gyun Park, Joon-Ho Lee.

**Formal analysis:** Young-gyun Park.

**Funding acquisition:** Joon-Ho Lee.

**Investigation:** Young-gyun Park, Joon-Ho Lee.

**Methodology:** Young-gyun Park.

**Project administration:** Joon-Ho Lee.

**Resources:** Joon-Ho Lee.

**Software:** Young-gyun Park.

**Supervision:** Joon-Ho Lee.

**Validation:** Young-gyun Park, Joon-Ho Lee.

**Visualization:** Young-gyun Park.

**Writing – original draft:** Young-gyun Park.

**Writing – review & editing:** Young-gyun Park, Joon-Ho Lee.

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
