## [Decision Letter · Decision Letter 0]

26 Oct 2020

PONE-D-20-30203

Increase of control efficacy of Nesidiocoris tenuis (Hemiptera: Miridae) in the greenhouse by enhancing its establishment using UV-LED light

PLOS ONE

Dear Dr. Lee,

Thank you for submitting your manuscript to PLOS ONE. After careful consideration, we feel that it has merit but does not fully meet PLOS ONE’s publication criteria as it currently stands. Therefore, we invite you to submit a revised version of the manuscript that addresses the points raised during the review process.

We look forward to receiving your revised manuscript.

Kind regards,

Antonio Biondi, Ph.D.

Academic Editor

PLOS ONE

Additional Editor Comments:

Your manuscript has been reviewed by three qualified referees. All of them were very critic with the manuscript quality, one recommended rejection, one was unsure whether rejecting or suggesting major revision and another thought that it is needed to read a revised version of it before giving a final decision. I actually agree with the latter. Therefore I invite the authors to fully follow the reviewer comments while preparing the new, deeply revised, manuscript version.

Particular attention should be paid on the field data design (too preliminary and with not well designed experiments), on the writing style (not suitable for an international journal), reference and topic considerations.

Finally, I am not sure you made all raw data underlying the findings in your manuscript fully available, please read the PLOS Data policy and add supplementary electronic material if needed.

Journal Requirements:

Reviewers' comments:

Reviewer's Responses to Questions

**Comments to the Author**

1. Is the manuscript technically sound, and do the data support the conclusions?

Reviewer #1: Partly

Reviewer #2: Partly

Reviewer #3: Partly

2. Has the statistical analysis been performed appropriately and rigorously? 

Reviewer #1: No

Reviewer #2: Yes

Reviewer #3: N/A

3. Have the authors made all data underlying the findings in their manuscript fully available?

Reviewer #1: No

Reviewer #2: No

Reviewer #3: Yes

4. Is the manuscript presented in an intelligible fashion and written in standard English?

Reviewer #1: No

Reviewer #2: No

Reviewer #3: No

5. Review Comments to the Author

Reviewer #1: Major comments

The manuscript intituled “Increase of control efficacy of Nesidiocoris tenuis (Hemiptera: Miridae) in the greenhouse by enhancing its establishment using UV-LED light” presents new finds concerning B. tabaci control by using N. tenuis and several kind of wavelengths. The introduction section should be enhanced, specifically concerning the biological traits of B. tabaci and N. tenuis. The objective of this work need be better explained. The manuscript presents several parts without link (i.e. i. why the authors used less insects in the experiment 2? ii. Why it was choose the predator wavelengths preference in greenhouse experiment? Etc). Additionally, the manuscript is poorly referenced. Finally, the whole manuscript ought to be revised by a native speaker.

Minor comments

L16: Replace “is a biological control agent for controlling” to “ is one of the biological control agents against..”

L17: I suggest to use: “The successful..”, “pests..”, and “for effective..”

L19-21: Why the authors want to attract B. tabaci for crops? In fact, you should attract natural enemies and repel pests. You need to write the goals more clearly.

L21: “LED light at a wavelength..”

L21-22: Incomplete statement, revise.

L27: “at a wavelength..”

L32: B. tabaci is a polyphagous pest, as you showed in your text; it appears that B. tabaci is a problem only for the cultures cited. Revise!

L38-39: Explain better about alternatives to control B. tabaci.

References suggested:

1. FEKRI, Masoomeh Samareh et al. The combined effect of some plant extracts and pesticide Pymetrozine and two tomato varieties on biological characteristics of Bemisia tabaci (Homoptera: Aleyrodidae) in greenhouse conditions. Entomologia Generalis, p. 229-242, 2016.

2. SOARES, Marianne A. et al. Detrimental sublethal effects hamper the effective use of natural and chemical pesticides in combination with a key natural enemy of Bemisia tabaci on tomato. Pest Management Science, 2020.

3. DONG, Yong‐Cheng et al. Nitrogen and water inputs to tomato plant do not trigger bottom‐up effects on a leafminer parasitoid through host and non‐host exposures. Pest management science, v. 74, n. 3, p. 516-522, 2018.

4. LYKOURESSIS, Dionyssios P. et al. Predation rates of Macrolophus pygmaeus (Hemiptera: Miridae) on different densities of eggs and nymphal instars of the greenhouse whitefly trialeurodes vaporariorum (Homoptera: Aleyrodidae). Entomologia generalis, v. 32, n. 2, p. 105-112, 2009.

L44-45: References suggested about the control of pests provided by N. tenuis.

1. BIONDI, Antonio et al. Can alternative host plant and prey affect phytophagy and biological control by the zoophytophagous mirid Nesidiocoris tenuis?. BioControl, v. 61, n. 1, p. 79-90, 2016.

2. CAMPOS, Mateus R. et al. From the Western Palaearctic region to beyond: Tuta absoluta 10 years after invading Europe. Journal of Pest Science, v. 90, n. 3, p. 787-796, 2017.

3. BIONDI, Antonio et al. Ecology, worldwide spread, and management of the invasive South American tomato pinworm, Tuta absoluta: past, present, and future. Annual Review of Entomology, v. 63, p. 239-258, 2018.

4. SOARES, Marianne A. et al. Botanical insecticide and natural enemies: a potential combination for pest management against Tuta absoluta. Journal of Pest Science, v. 92, n. 4, p. 1433-1443, 2019.

5. DESNEUX, Nicolas et al. Biological invasion of European tomato crops by Tuta absoluta: ecology, geographic expansion and prospects for biological control. Journal of pest science, v. 83, n. 3, p. 197-215, 2010.

L53: Replace “settlement” to “establishment”

L70: As I told before, the objectives should be clearer.

L82: What does meaning “for ca. 1 h”? Unclear!

” Figure legend: Provide all legends in a single page. Consequently, you should remove these legends from the continuous text.

L84: Remove both “(S.D.)”

L90: Replace “B. tabaci” to “Bemisia tabaci”. After dot all scientific names should write unabbreviated

L94: Did you observe cannibalism between nymphs and adults?

L96: I totally disagree with all title of the experiments. In fact, the title is a brief of each experiment. Revise all!

L98: “To find the proper”

L110: Male or female? Age? Provide in the manuscript.

L11: Did you use a yellow trap? Provide in the manuscript.

L113-115: Move to “statistical analysis” section

L122: Remove both “(S.D.)”

L127: How many insects were used in each replication?

L131: Replace “petri” to “Petri”. Revise the complete manuscript.

L135: Replace “to provide refuge” to “to provide refuge and food substrate”

L135: Replace “also” to “additionally”

L163-165: Move to “statistical analysis” section

L169-176: In my opinion, this part is useless. You can just mention that this preliminary test was carried out before the real experiment.

L200-202: You can omit this information.

L210: Why in the experiment 2 less insects than the experiment 1 were released?

L218-227: You should provide this information in your table. Think to optimize how to show the information in your manuscript.

L227: Replace “Between” to “among

L230-231: Use the scientific language.

L231-233: There is no sense in this affirmation. Revise to use the correct statistical analysis! Additionally, You should move this part to statistical analysis section.

L239-240: It seems results! Revise!

L242-248: You should provide this information in your table.

Table 3 – The statistical analysis in no response results looks wrong. Are you sure that all is letter a?

L257-258: It seems discussion.

L279-287: These results could fit well in one table.

L411-415: Very confusing! Why it was choose the predator preference?

L449-450: Explain better.

L460-462: Reference

Reviewer #2: I read this interesting paper with pleasure. A part from some minor needed corrections (mostly on writing, citations and small details) pointed out in the specific comments below, I have a very major concern on the field assay. In that trial authors did not ensure the isolation of the crop with the environment around, this led major doubts on the obtained results. Moreover, in the first attempt they released a very few predators that did not allow to get reliable results. Finally, a varying number of insect pest was used among the two field trials. Unfortunately, all of this does allow to have a sound experimental setup. Therefore, my best suggestion is to delete (from all relevant sections, included the title) the greenhouse part of this manuscript.

L16: I suggest this first sentence: “The zoophytophagous mirid bug Nesidiocoris tenuis is a biological control agent of the whitefly Bemisia tabaci……”

L18: “the” may be added between “with” and “pest”

L24: which wavelength? The same? May be “this wavelength” or “the same wavelength”?

L28-29: I suggest: “in early establishment of this predatory mirid bug,…” “and proper control of the whitefly”

L33: add “on” before “them”

L35: add “the” before “tomato”

L38: change “B. tabaci” to “this insect pest”. Please revise the whole manuscript for this. Indeed, it is not nice and elegant writing so many times the same word, even though is a species names.

L38: add “strategies” between “control” and “has”

L43: add “pest” before “biological control”

L45: here it could be good if authors cite also two major and recent review articles in which the role of N. tenuis on T. absoluta predations is emphasized (doi: 10.1146/annurev-ento-031616-034933; doi: 10.1127/entomologia/2018/0749)

L49-52: I suggest authors to include here a statement about the study of Naselli et al. 2017 (https://doi.org/10.1007/s11829-016-9481-5) who evaluated the potential of plant volatiles belonging from alternative host plants as useful tool for managing the mirid behaviour and potentially reduce its feeding activity on cultivated (i.e. tomato) plants..

L54: citations on prey and alternative prey availability need to be included (e.g., DOI 10.1007/s10526-015-9700-5. Doi: 10.1127/entomologia/2019/0824). Moreover, s the role of alternative and banker plants would need to be at least mentioned here as well (e.g., DOI: 10.1127/entomologia/2019/0625; doi: DOI: 10.1080/09670874.2012.659229)

L56-58: confusing statements. Please rephrase

L67: change “establishment of N. tenuis” to “establishment of this predatory mirid bug”

L68: change “help control pests such as B. tabaci” to “help pest (i.e., B. tabaci) control”

L68-69: In my opinion the sentence “In addition, ….. agent.” Is repetitive and not necessary here. I suggest deleting it.

L74-75: as currently written seems that “enhancing spatial coherence … could successfully control B. tabaci”. I think a verb is needed between “successfully” and “control” (maybe improve, ameliorate, etc). Alternatively sentence modifications should be done to make sense to the statement.

L77: change “Methods” to “methods”

L78: “Test insects” does not sound good to me. Please provide a better subtitle

L82. Spell out the whole genus name after “use.”

L82: Please specify if the commercial N. tenuis specimens were used only to start the laboratory rearing. It is not clear in the current sentence text.

L89: change “This” to “These” or start the sentence with some words like “Newly obtained N. tenuis adults…” or similar.

L90: see the comment of line 82

L92: provide mean values ± S.D. or S.E. if possible

L100-101: provide specifications about position, size, orientation etc. of the transparent sticky trap inside Y-tube branches

L102: how these LED light were applied (position, distance, lighted surface)? please specify

L109: specify if 30 specimens were used for each replication (L106) (for a total of 150 specimens per tested wavelength) or in total. It is not completely clear.

L110-111: 50 B. tabaci specimens per tested wavelength in total? had you replications? Please provide more details about this relevant trial aspect

L117-118: confusing title, please rephrase

L178: the “Experimental design and sampling method” section is too long and sometimes redundant. Statement such as “Most UV-C from the sun is absorbed into the ozone layer.” could be deleted. General contents should focus on relevant aspects. I suggest rephrasing most of this section by writing concisely and linearly. Sentences of lines 202-206 should come before those of lines 199-202.

L228: you stated “June 19, the last release date of N. tenuis” but the last release was “on June 14”. Please verify and/or correct

L263-265: I suggest deleting percentage values from the statement “Five wavelengths attracted … > 405 nm (41.6%).” because it is repetitive. Indeed, reported results occur in table 2 and some in lines 261-262. (may be better: “385 nm > 365 nm > 445 ………”)

L268: I suggest adding a specification about the statement “except for wavelength at 445 nm”. As it is currently written, it could seem that 445 nm was the most attractive while it was the least attractive. Please modify or provide specifications

L330-332: P values do not confirm that “not significantly different”. Verify and clarify

L337-339: this should be moved to the discussion

L354-355: “seemed to help the control the activity of N. tenuis”. Provide correction

L357-358: these statements should be moved to the discussion

L390-391: this statement should be moved to the discussion

L411-412: spell out the genus after the point “preference. N.”

L452-454: confusing sentence. Rephrase

Reviewer #3: The manuscript “ Increase of control efficacy of Nesidiocoris tenuis (Hemiptera: Miridae) in the

greenhouse by enhancing its establishment using UV-LED light”, study the attraction and effect of Led Light on Nesidiocoris tenuis and the herbivore Bemisia tabaci, under laboratory and greenhouse conditions. In general, the objective of the work seems interesting to me. However, I have doubts about recommending publishing this paper in PlosOne.

In general, I found some problems with this manuscript in terms of title, the purpose of study, experiments, and results from interpretation. Thus, I really have some concerns about this work, which could be a problem to publish in this journal.

At least for this reviewer, the ms needs a review of the English language. Furthermore, some wordings are challenging to understand ("spatial coherence", “settlement”, "spatial separation", etc…).

Some of the main concerns and problems are as follows:

Title:

The title is tough to understand, at least for this reviewer. I suggest to try with something like this: “LED lights enhance the establishment and biological control efficacy of N. tenuis”.

Introduction:

Authors introduce both insects well; however, at least for this reviewer, the introduction section should have a better hypothesis of the work and made the objectives clear.

Line 40-42: This paragraph should move below after to introduce N. tenuis.

Line 50-52: There are also alternative food sources as sugars that reduce this damage in tomato plants, authors should be a reference to these works...

Line 59: There are also alternative food sources as sugars that improve the establishment of N. tenuis in tomato plants; authors should be a reference to these works...

Line 66: Authors should make the hypothesis of the study more clear

Line 72: Why the authors want to attract B. tabaci to tomato plants?? Could this attraction be a detriment for N. tenuis biocontrol?

M&M:

Line 81: “25ºC” needs (± SE). Here and throughout the manuscript.

Line 82: “at 25ºC for ca. 1h before use”, at least for this reviewer, this is a little bit misleading if these N. tenuis are for rearing.

Line 88: Why the authors use eggs of C. cautella to evaluate the predation rate instead of other eggs source (e.g. Tuta absoluta, Ephestia kuehniella)?

Line 89: Which N.tenuis? How old are N. tenuis used in ALL the experiments? Are males or females?

Line 96: Although the Y-tube results seem that the response of B. tabaci and N. tenuis are reliable enough. I think that the methodology of this experiment needs to be more explicit. How the authors separated the “lighting zone”? Is there any work that used this methodology for light attraction?

Line 100: “branches” means “arms”?

Line 108: “About 30 N. tenuis”?. The authors need to be more concise.

Line 110: “Approximately 50 B. tabaci”?. The authors need to be more concise.

Line 117: this section evaluates the predation rate. It is okay. However, how the authors evaluate the establishment?

Line 152: Which is the second experiment?

Line 198: N. tenuis population includes nymphs and adults? Also, for the predation experiment?

Line 199: all the tomato plant leaves were counted??

Line 203-204: “7 to 20 N. tenuis were release”? The number of N. tenuis released is essential for results interpretation.

Line 204-205: same for the number of B. tabaci

Line 221: Calvo et al. 2012 showed that 0.25-0.5 N. tenuis per plant was a good release range.

Results:

Line 258: Removed this sentence

Table 2: It is known that results from Y-tube should be shown in vertical bars figures. For this reason, at least for this reviewer, results from the Y-tube will be better represented by figures (e.g. left side responders to different LED light treatments, right side responders to control treatment, and outside the figure but next to it, the number of individuals who did not respond should be included).

Greenhouse experiments:

Why did the authors not use a GLMM to analyze the greenhouse data?? This model can include factors as the date and interaction date*treatment…

Line 354: “In N. tenuis released plots, 385 nm wavelength LED light seemed to help the control the activity of N. tenuis for B. tabaci”. However, when you showed the figure, there are no differences between treatments in B. tabaci populations…It seems that LED lights did not affect B. tabaci control by N. tenuis.

Line 368: At least for this reviewer, it is hard to understand how the percentage of control value of B. tabaci + N. tenuis + LED are higher in all the sample dates if the B. tabaci levels are not significantly different.

Line 371: The experiment 2 showed that LED lights' effect on B. tabaci populations reductions correlated with N. tenuis populations increases. The authors should justify better the results of the first experiment throughout the discussion.

Discussion

In my consideration, the discussion should be rewritten again. Authors need to be more consistent with their results and then discuss better with other works. There are a large number of sentences that need references.

Line 421 -422 and Line 433-434 are almost the same sentences. In my opinion, I removed the first one because this sentence here did not match the flow of the discussion.

Line 439 and line 443: “supported results of the greenhouse experiment” (experiment 1 or 2?)

Line 450: Figure 8. From where are these temperature and humidity data? Which greenhouse? If authors represent each day's abiotic conditions, they should put the temperature, and RH mean and SE.

Line 468-469: Experiment 1 or 2?

6. PLOS authors have the option to publish the peer review history of their article (what does this mean?). If published, this will include your full peer review and any attached files.

Reviewer #1: No

Reviewer #2: No

Reviewer #3: No

---

## [Author Response · Author response to Decision Letter 0]

29 Nov 2020

Dear Editor

Thank you for giving us another chance and the helpful comments on our manuscript. We have faithfully revised our manuscript by referring to the reviewer's opinion. And, according to the reviewer’s suggestion, our manuscript was professionally re-edited for English by "VanDriesche Scientific Editing". The address of Young-gyun Park was changed, thus we added: “Present address” on our title page. Also, we newly upload our figures with better resolution (Fig 8 was revised according to the reviewer’s comment). Moreover, according to the opinion of Reviewer 3, we changed the title to "UV-LED Lights Enhance the Establishment and Biological Control Efficacy of Nesidiocoris tenuis (Reuter) (Hemiptera: Miridae) in the Greenhouse".

Revised parts for reviewer 1, 2, and 3 were marked in red, green, and orange color in the text, respectively. 

Reviewer’s comments and our responses (Reviewer #1)

Major comments

The manuscript intitled “Increase of control efficacy of Nesidiocoris tenuis (Hemiptera: Miridae) in the greenhouse by enhancing its establishment using UV-LED light” presents new finds concerning B. tabaci control by using N. tenuis and several kind of wavelengths. The introduction section should be enhanced, specifically concerning the biological traits of B. tabaci and N. tenuis. The objective of this work need be better explained. The manuscript presents several parts without link (i.e. i. why the authors used less insects in the experiment 2? ii. Why it was chosen the predator wavelengths preference in greenhouse experiment? Etc). Additionally, the manuscript is poorly referenced. Finally, the whole manuscript ought to be revised by a native speaker.

Our response: Thank you for the comments. We sincerely appreciate your suggestions. We revised our manuscript by referring to your suggestion. Your questions in major comments were answered in the minor comments section. And, according to the reviewer’s suggestion, our manuscript was professionally re-edited for English by "VanDriesche Scientific Editing"

Minor comments

1. Reviewer comment: L16: Replace “is a biological control agent for controlling” to “ is one of the biological control agents against..”

Our response: We changed it according to the suggestion. Thank you. Please check L18-19 in the revised manuscript.

2. Reviewer comment: L17: I suggest to use: “The successful..”, “pests..”, and “for effective..”

Our response: We changed it according to the suggestion and English editing process. Thank you. Please check L20-22 in the revised manuscript.

Original sentence: Successful establishment of a biological control agent and its spatial coherence with pest in the target area are essential for an effective biological control in greenhouses.

Revised sentence: The successful establishment of a biological control agent and co-occurrence with the target pests increases the efficacy of biological control in greenhouses.

3. Reviewer comment: L19-21: Why the authors want to attract B. tabaci for crops? In fact, you should attract natural enemies and repel pests. You need to write the goals more clearly.

Our response: Thank you for the nice question. However, unlike pheromone that can attract specific species, it is very tough to induce attraction response to one species using a certain wavelength of light with high efficiency. There were many reports about the positive response for UV or near-UV for pests and natural enemies. Some wavelengths of those ranges were finally chosen for enhancing the establishment of natural enemies [1-3]. In these results, applying a certain wavelength to attract specific species of insects might be hard. Thus, it would be better to consider the spatial congruence of both pests and natural enemies for enhancing the efficacy of biological control.

 The statement in the abstract section in the original manuscript may cause some confusion, as the reviewer raised. Thus, we revised that sentence. Please check L22-24 in the revised manuscript.

Original sentence: In this study, we explored proper wavelengths of LED light that could attract both B. tabaci and N. tenuis and enhance their spatial coherence so that biological control of B. tabaci could be more effective.

Revised sentence: In this study, we explored the effects of different wavelengths of LED light on establishment of N. tenuis, with the goal of enhancing the biological control of B. tabaci in greenhouse crop. 

References

1. Johansen NS, Vänninen I, Pinto DM, Nissinen AI, Shipp L. In the light of new greenhouse technologies: 2. Direct effects of artificial lighting on arthropods and integrated pest management in greenhouse crops. Ann Appl Biol. 2011; 159(1): 1-27.

2. Ogino T, Uehara T, Muraji M, Yamaguchi T, Ichihashi T, Suzuki T, et al. Violet LED light enhances the recruitment of a thrip predator in open fields. Sci Rep. 2016; 6: 32302.

3. Uehara T, Ogino T, Nakano A, Tezuka T, Yamaguchi T, Kainoh Y, et al. Violet light is the most effective wavelength for recruiting the predatory bug Nesidiocoris tenuis. BioControl. 2019; 64(2): 139-147.

4. Reviewer comment: L21: “LED light at a wavelength..”

Our response: We changed it according to the suggestion. Thank you. Please check L24-25 in the revised manuscript.

5. Reviewer comment: L21-22: Incomplete statement, revise.

Our response: We changed it. Thank you. Please check L24-25 in revised manuscript.

Original sentence: LED light at a wavelength of 385 nm attracted N. tenuis the most.

Revised sentence: Nesidiocoris tenuis was most strongly attracted by LED light at a wavelength of 385 nm.

6. Reviewer comment: L27: “at a wavelength..”

Our response: We changed it according to the suggestion. Also, we found same error in L25, and it was corrected. Please check L29 and 32 in the revised manuscript.

7. Reviewer comment: L32: B. tabaci is a polyphagous pest, as you showed in your text; it appears that B. tabaci is a problem only for the cultures cited. Revise!

Our response: We changed “greenhouse” to “horticultural” and added “etc.” after “flower”. Please check L38 in the revised manuscript.

8. Reviewer comment: L38-39: Explain better about alternatives to control B. tabaci.

References suggested:

1. FEKRI, Masoomeh Samareh et al. The combined effect of some plant extracts and pesticide Pymetrozine and two tomato varieties on biological characteristics of Bemisia tabaci (Homoptera: Aleyrodidae) in greenhouse conditions. Entomologia Generalis, p. 229-242, 2016.

2. SOARES, Marianne A. et al. Detrimental sublethal effects hamper the effective use of natural and chemical pesticides in combination with a key natural enemy of Bemisia tabaci on tomato. Pest Management Science, 2020.

3. DONG, Yong‐Cheng et al. Nitrogen and water inputs to tomato plant do not trigger bottom‐up effects on a leafminer parasitoid through host and non‐host exposures. Pest management science, v. 74, n. 3, p. 516-522, 2018.

4. LYKOURESSIS, Dionyssios P. et al. Predation rates of Macrolophus pygmaeus (Hemiptera: Miridae) on different densities of eggs and nymphal instars of the greenhouse whitefly Trialeurodes vaporariorum (Homoptera: Aleyrodidae). Entomologia generalis, v. 32, n. 2, p. 105-112, 2009.

Our response: Thank you for the brilliant suggestion. We changed it according to the suggestion with references. However, your suggestion's 4th reference was omitted in our revision because it was a study for Trialeurodes vaporariorum, not Bemisia tabaci. Please check L44-47, 557, 564, and 568 in the revised manuscript.

Original sentence: Thus, biological control has been increasingly promoted for the control of B. tabaci.

Revised sentence: Therefore, alternative control strategies such as combinations of pesticides and plant extracts, manipulations of tri-trophic effects, and biological control have become of increasing interest for control of B. tabaci. 

9. Reviewer comment: L44-45: References suggested about the control of pests provided by N. tenuis.

1. BIONDI, Antonio et al. Can alternative host plant and prey affect phytophagy and biological control by the zoophytophagous mirid Nesidiocoris tenuis?. BioControl, v. 61, n. 1, p. 79-90, 2016.

2. CAMPOS, Mateus R. et al. From the Western Palaearctic region to beyond: Tuta absoluta 10 years after invading Europe. Journal of Pest Science, v. 90, n. 3, p. 787-796, 2017.

3. BIONDI, Antonio et al. Ecology, worldwide spread, and management of the invasive South American tomato pinworm, Tuta absoluta: past, present, and future. Annual Review of Entomology, v. 63, p. 239-258, 2018.

4. SOARES, Marianne A. et al. Botanical insecticide and natural enemies: a potential combination for pest management against Tuta absoluta. Journal of Pest Science, v. 92, n. 4, p. 1433-1443, 2019.

5. DESNEUX, Nicolas et al. Biological invasion of European tomato crops by Tuta absoluta: ecology, geographic expansion and prospects for biological control. Journal of pest science, v. 83, n. 3, p. 197-215, 2010.

Our response: Thank you for the brilliant suggestion. We added the suggested references in our manuscript. Please check L50-51, 586, 600, 603, 610, and 613 in the revised manuscript.

10. Reviewer comment: L53: Replace “settlement” to “establishment”

Our response: We changed it according to the suggestion. Please check L63 in the revised manuscript.

11. Reviewer comment: L70: As I told before, the objectives should be clearer.

Our response: Thank you for the suggestion. We revised our sentences. Please check L86-95 in the revised manuscript.

Original sentences: If a selected LED light shows asymmetrical response between N. tenuis and its prey, it could result in spatial separation between the predator and the prey. Therefore, the objective of this study was to explore the proper wavelength of LED light to attract both B. tabaci and its predator, N. tenuis, and determine if enhancing spatial coherence of these two species using LED light could successfully control B. tabaci in a tomato greenhouse.

Revised sentences: If a selected LED light is asymmetrically attractive between N. tenuis and its prey, the spatial separation between the predator and the prey, rather than being reduced, might increase. Therefore, we hypothesized that the best strategy would be to select LED light wavelengths that have high attractiveness for both the predator and prey. The specific objectives of this study, therefore, were to determine if selected wavelengths of LED light would enhance establishment of N. tenuis and thus improve biocontrol of B. tabaci in a tomato greenhouse. Additionally, the responses of B. tabaci to selected wavelengths of LED light were explored to determine their impact on the spatial congruence of the distributions of natural enemies and the target pest.

12. Reviewer comment: L82: What does meaning “for ca. 1 h”? Unclear!

Our response: We used “ca.” as a mean of “about”. So, we changed it to “about”. Please check L102 in the revised manuscript.

13. Reviewer comment: Figure legend: Provide all legends in a single page. Consequently, you should remove these legends from the continuous text.

Our response: We followed the formatting guidelines of PLoS ONE (https://journals.plos.org/plosone/s/file?id=wjVg/PLOSOne_formatting_sample_main_body.pdf%20).

We followed the guideline below.

14. Reviewer comment: L84: Remove both “(S.D.)”

Our response: Remove done. Please check L105 in the revised manuscript.

15. Reviewer comment: L90: Replace “B. tabaci” to “Bemisia tabaci”. After dot all scientific names should write unabbreviated

Our response: We changed it. Please check L110 in the revised manuscript.

16. Reviewer comment: L94: Did you observe cannibalism between nymphs and adults?

Our response: We gave enough eggs of C. cautella as food for N. tenuis. We observed so many times when we changed the food or just wanted to observe, and the nymphs and adults always got along well. 

17. Reviewer comment: L96: I totally disagree with all title of the experiments. In fact, the title is a brief of each experiment. Revise all!

Our response: Revision was done.

Test insects -> Insects sources and rearing

Y-tube experiments -> Exp. #1: Wavelength selection (in Y-tube tests)

Incubator experiment for verification of enhanced establishment rate 

-> This part was divided into two chapters.

-> Exp. #2: No-choice test of effect of light treatment on N. tenuis movement and predator efficiency 

-> Exp. #3: Choice test of effect of light treatment on N. tenuis movement and predator efficiency 

Greenhouse experiment 

-> This part was divided into two chapters.

-> Exp. #4: Effects of selected wavelengths on predator and prey in a greenhouse trial

-> Exp. #5: Second tomato greenhouse trial

Please check L98, 117, 146, 204, 214, 267, 296, 321, 350, and 400 in the revised manuscript.

18. Reviewer comment: L98: “To find the proper”

Our response: We revised the whole of the sentence with English editing. Please check L118-120 in the revised manuscript.

Original sentence: Y-tube (200 mm in length and 40 mm in diameter for each branch) tests were conducted to find proper wavelength of LED light that would shows the highest attraction rate for N. tenuis and B. tabaci.

Revised sentence: Wavelength selection experiments were conducted to find the wavelength of LED light that would attract both N. tenuis and B. tabaci by using a Y-tube assay (200 mm in length and 40 mm in diameter for each branch).

19. Reviewer comment: L110: Male or female? Age? Provide in the manuscript.

Our response: We concentrated on the response of the group; thus, we randomly collected from the colony. We added “randomly selected from the purchased bottle” for N. tenuis and “randomly selected from the rearing colony” for B. tabaci. Please check L135-136 and L138-139 in the revised manuscript.

20. Reviewer comment: L111: Did you use a yellow trap? Provide in the manuscript.

Our response: No, we used the transparent sticky traps for Y-tube test. Actually, we already mentioned it in the same section. However, we added “transparent” according to your suggestion. Please check L140 in the revised manuscript.

21. Reviewer comment: L113-115: Move to “statistical analysis” section

Our response: Move done. Please check L280-282 in the revised manuscript.

22. Reviewer comment: L122: Remove both “(S.D.)”

Our response: Remove done. Please check L156 in the revised manuscript.

23. Reviewer comment: L127: How many insects were used in each replication?

Our response: Ten N. tenuis were used for the first experiment (Exp. #2) in an incubator. And, 15 N. tenuis were used for the second experiment (Exp. #3) in an incubator. We mentioned this in our manuscript. Please check L186 and 211 in the revised manuscript.

24. Reviewer comment: L131: Replace “petri” to “Petri”. Revise the complete manuscript.

Our response: We changed all in our manuscript. Please check L165, 166, 171, 172, 187, 189, and 193 in the revised manuscript.

25. Reviewer comment: L135: Replace “to provide refuge” to “to provide refuge and food substrate”

Our response: Thank you. correction was done. Please check L169 in the revised manuscript.

26. Reviewer comment: L135: Replace “also” to “additionally”

Our response: We revised the whole of the sentence with English editing. Please check L169-173 in the revised manuscript.

Original sentence: Also, 100 eggs of C. cautella on a small Petri dish (40 mm x 6 mm; diameter x height) were provided as food in each large Petri dish (100 mm x 42 mm; diameter x height).

Revised sentence: To measure predation, feeding on a non-pest was used as a predation index to compare treatments. This index was the rate of predation on 100 eggs of C. cautella, which were placed on a small Petri dish (40 mm x 6 mm; diameter x height), which itself was placed on the large Petri dish (100 mm x 42 mm; diameter x height) on which the tomato stems had been placed.

27. Reviewer comment: L163-165: Move to “statistical analysis” section

Our response: Move done. Please check L283-286 in the revised manuscript.

28. Reviewer comment: L169-176: In my opinion, this part is useless. You can just mention that this preliminary test was carried out before the real experiment.

Our response: We deleted that part in M & M and the results. And, we added a relevant sentence in M & M of the incubator experiment (Exp. #2 and #3). And also, the figure legend was moved into that part and. In the figure legend, we added information about the test method. Please check L179-180, 199-202, 214 in the revised manuscript.

Added sentence in M & M part of incubator experiment: The timing of when the additional lighting occurrence was based on data from a preliminary test in a greenhouse (Fig 1). 

Figure legends: Fig 1. Number (Mean ± S.E.) of N. tenuis adults attracted to 385 nm wavelength LED light (9-W) developed in a greenhouse (14.5 x 7 m2; width x length) without plants (three replications, 100 N. tenuis in each replication) at different points in the daily light cycle.

29. Reviewer comment: L200-202: You can omit this information.

Our response: Correction was done. Please check L243 in the revised manuscript.

30. Reviewer comment: L210: Why in the experiment 2 less insects than the experiment 1 were released?

Our response: Thank you for the question. As we mentioned in the discussion, the initial environmental conditions of the Exp. #4 (revised version of experiment 1) were unsuitable for both B. tabaci and N. tenuis. Actually, no heating system was operated, unlike real commercial cultural greenhouses. Densities of both insects in all treatment plots were consistently low until June 12. Thus, we released N. tenuis and B. tabaci several times. On the contrary, after transplanting of tomato plants to the greenhouse in the Exp. #5 (revised version of experiment 2), the external inflow of B. tabaci (5 to 10 per plant) occurred. Thus, we released less number of insects in the Exp. #5.

31. Reviewer comment: L218-227: You should provide this information in your table. Think to optimize how to show the information in your manuscript. 

Our response: We added the content of L218-227 in Table 1, and revised that part. Please check L246 and 253-257 in revised manuscript.

Original sentences: First, 100 adults of B. tabaci were released to all plots on April 24 and 150 adults of B. tabaci were additionally released on April 30 before the first observation day. Then seven adults of N. tenuis were released to <B. tabaci + N. tenuis> and <B. tabaci + N. tenuis + LED> plots on May 9. The optimal release number of N. tenuis is known as one individual per plant. However, we released 0.5 N. tenuis per plant instead due to a lower density of B. tabaci in our experimental plots. For the next release, seven adults of N. tenuis were released to <B. tabaci + N. tenuis> and <B. tabaci + N. tenuis + LED> plots on May 21 and 100 adults of B. tabaci were released to all plots on May 30. Lastly, 14 adults of N. tenuis were released to <B. tabaci + N. tenuis> and <B. tabaci + N. tenuis + LED> plots on June 14.

Revised sentences: The optimal release number of N. tenuis is known as one individual per plant [30]. However, in the 1st and 2nd predator releases, we used only 0.5 N. tenuis per plant due to the low density of B. tabaci in our experimental plots. In the last release of N. tenuis, one bug per plant was released because the density of mirid bugs at that time was low. 

32. Reviewer comment: L227: Replace “Between” to “among

Our response: Replacement was done. Please check L257 in the revised manuscript.

33. Reviewer comment: L230-231: Use the scientific language.

Our response: Revision was done. Please check L260-261 in the revised manuscript.

34. Reviewer comment: L231-233: There is no sense in this affirmation. Revise to use the correct statistical analysis! Additionally, You should move this part to statistical analysis section.

Our response: Thank you. Before June 14, the insect densities of all plots were almost zero. And, the release of the insect was not finished until that time. Thus we conducted the statistical analysis after that time. We revised the sentence, and it was moved to the data analysis section. Please check L287-293 in revised manuscript.

Original sentence: Statistical analysis of weekly density was also performed from June 19 because densities of B. tabaci and N. tenuis before June 19 were too low to obtain significant results.

Revised sentence: In Exp. 4, analysis of the weekly densities of predator and prey were performed from starting on June 19, the first observation date after the last release date (June 14) of N. tenuis, because densities of B. tabaci and N. tenuis before June 19 were too low for analysis.

35. Reviewer comment: L239-240: It seems results! Revise!

Our response: Thank you. We revised the sentence. Please check L268-269 in the revised manuscript.

Original sentence: In experiment 1, the effect of 385 nm wavelength on the density of N. tenuis was not significant. 

Revised sentences: In the Exp. #5, we assessed the effect of the presence or absence of LED light (385 nm) on predation and control of B. tabaci by N. tenuis. 

35. Reviewer comment: L242-248: You should provide this information in your table.

Our response: We added the information about the released plot in Table 1. And, we deleted the content of the release date and plot in L242-248 of the original manuscript. However, we think that the remaining other part is better. Please check L270-277 in the revised manuscript.

Original sentences: All plots (LED or non-LED) received adults of both B. tabaci and N. tenuis. Thus, each treatment had six replications in three greenhouses. Tomato seedlings (about 300 mm in height) were planted on August 28 and 100 B. tabaci adults were released to each plot on Aug 29. Before releasing B. tabaci, five to ten B. tabaci adults per plant were already present in each plant. Thus, we released 20 N. tenuis adults to each plot on September 3. Weekly mean densities of B. tabaci and N. tenuis were compared between non-LED and LED treatments. 

Revised sentences: There were two treatments (LED or non-LED light) and all four plots in each of the three greenhouses received adults of both B. tabaci and N. tenuis (Table 1). Thus, each treatment had six total replications, two in each of the three greenhouses. Tomato seedlings (about 30 cm in height) were planted on August 28. Before our first release of B. tabaci, we observed that there were five to ten adult whiteflies per plant already on the tomato plants. Thus, we released 20 N. tenuis adults in each plot to achieve the desired predator/prey ratio. Weekly mean densities of B. tabaci and N. tenuis were compared between non-LED and LED treatments. 

36. Reviewer comment: Table 3 – The statistical analysis in no response results looks wrong. Are you sure that all is letter a?

Our response: Yes, the statistical values of the no response rate in B. tabaci are here.

F4, 20 = 2.27, P = 0.0977

And, this graph is distribution of rate made by PROC ANOVA in SAS.

Thus, the non-significant result of no response rate in B. tabaci seems to be occurred by high variation of data.

37. Reviewer comment: L257-258: It seems discussion.

Our response: 

The line 257-258 you mentioned is probably lines 357-358 because L257-258 is simply a subheading ‘Y-tube experiment’ and a statement ‘Results of the Y-tube experiment for N. tenuis are presented in Table 2’ . 

 .

We think you probably intended L357-358.

The sentence of L357-358 is “It might be due to a later colonization of N. tenuis caused by an indeterminate reason such as temperature.”. We moved that sentence to our discussion with some revision (L481-483 in the revised manuscript).

Revised sentence: This lack of impact on the predator may have been due to the later colonization of N. tenuis of our plots due to low temperature or low whitefly density early in the trial.

38. Reviewer comment: L279-287: These results could fit well in one table.

Our response: Similarly, L279-287 might be L379-387. Thank you for the suggestion. We made Table 5 for those values, and we deleted those in the manuscript. Please check L408, 413 in the revised manuscript.

39. Reviewer comment: L411-415: Very confusing! Why it was choose the predator preference?

Our response: Thank you for the question. First of all, there was no significant difference between 365 and 385 nm wavelengths in both N. tenuis and B. tabaci. As previously mentioned in this response letter, it is tough to induce just one species' attraction response to certain wavelength light with high efficiency. Thus, we consider the spatial congruence of them, N. tenuis and B. tabaci. And our purpose of this study is to increase the control efficacy of natural enemy by enhancing establishment. Thus, we focused on the response of N. tenuis. Also, we thought that 365 nm wavelength light would be a proper wavelength for this strategy. Please check L82-95 in the revised manuscript. It is probably helpful.

40. Reviewer comment: L449-450: Explain better.

Our response: Thank you. We revised our sentence. Please check L486-487 in the revised manuscript.

Original sentence: One reason could be different environmental conditions.

Revised sentence: Differences in environmental conditions between the two greenhouse trials may have occurred. 

41. Reviewer comment: L460-462: Reference

Our response: Thank you. Actually, the reference for this sentence is our study, Fig 3. Thus, we added (Fig 3) at the end of the sentence. Please check L504 in the revised manuscript.

Reviewer’s comments and our responses (Reviewer #2)

1. Reviewer comment: I read this interesting paper with pleasure. A part from some minor needed corrections (mostly on writing, citations and small details) pointed out in the specific comments below, I have a very major concern on the field assay. In that trial authors did not ensure the isolation of the crop with the environment around, this led major doubts on the obtained results. Moreover, in the first attempt they released a very few predators that did not allow to get reliable results. Finally, a varying number of insect pest was used among the two field trials. Unfortunately, all of this does allow to have a sound experimental setup. Therefore, my best suggestion is to delete (from all relevant sections, included the title) the greenhouse part of this manuscript.

Our response: Thank you for the comments. We sincerely appreciate your suggestions. We revised our manuscript by referring to your suggestion. And, we realized that the English of our manuscript has a problem. Thus, our manuscript was professionally re-edited for English by "VanDriesche Scientific Editing"

Our test plots in greenhouse experiments were well isolated from the environment around. We missed the mention about the nylon screen in the window. Thus we added a sentence in our M & M section ("Nylon screening was also used to cover the vents."). Please check L226 in the revised manuscript.

We tried to form an environment as close to reality as possible. We think that you worried about the external inflow of B. tabaci in the Exp. #5 (revised version of experiment 2). But, complete blocking of B. tabaci inflow to the greenhouse is very hard. As you know, it is almost impossible. And this is a common occurrence in a real farmer's greenhouse. Of course, we can use a denser screen for blocking. However, in this case, ventilation capability would be decreased, potentially harmful to the plant.

2. Reviewer comment: L16: I suggest this first sentence: “The zoophytophagous mirid bug Nesidiocoris tenuis is a biological control agent of the whitefly Bemisia tabaci……”

Our response: Thank you for the suggestion. We changed our sentence according to your suggestion. Please check L18-19 in the revised manuscript.

3. Reviewer comment: L18: “the” may be added between “with” and “pest”

Our response: Thank you. We changed our sentence according to your suggestion. Please check L21 in the revised manuscript.

4. Reviewer comment: L24: which wavelength? The same? May be “this wavelength” or “the same wavelength”?

Our response: Thank you. We added “this” in front of “wavelength”. Please check L28 in the revised manuscript.

5. Reviewer comment: L28-29: I suggest: “in early establishment of this predatory mirid bug,…” “and proper control of the whitefly”

Our response: Thank you. We changed the sentence. Please check L33-35 in the revised manuscript.

6. Reviewer comment: L33: add “on” before “them”

Our response: Thank you. We added “on” before “them”. Please check L39 in the revised manuscript.

7. Reviewer comment: L35: add “the” before “tomato”

Our response: Thank you. We added “the” before “tomato”. Please check L41 in the revised manuscript.

8. Reviewer comment: L38: change “B. tabaci” to “this insect pest”. Please revise the whole manuscript for this. Indeed, it is not nice and elegant writing so many times the same word, even though is a species names.

Our response: Thank you. Revise done. And also, we revised the whole manuscript for this. Please check L44 in the revised manuscript.

9. Reviewer comment: L38: add “strategies” between “control” and “has”

Our response: Thank you. We already changed that sentence according to the suggestion of Reviewer 1. Thus, we added the “strategies” after “alternative control”. Please check L45 in the revised manuscript.

10. Reviewer comment: L43: add “pest” before “biological control”

Our response: Thank you. Add done. Please check L49 in the revised manuscript.

11. Reviewer comment: L45: here it could be good if authors cite also two major and recent review articles in which the role of N. tenuis on T. absoluta predations is emphasized (doi: 10.1146/annurev-ento-031616-034933; doi: 10.1127/entomologia/2018/0749)

Our response: Thank you for the suggestion. We cited suggested work in our manuscript. Please check L51, 606, and 610 in the revised manuscript.

12. Reviewer comment: L49-52: I suggest authors to include here a statement about the study of Naselli et al. 2017 (https://doi.org/10.1007/s11829-016-9481-5) who evaluated the potential of plant volatiles belonging from alternative host plants as useful tool for managing the mirid behaviour and potentially reduce its feeding activity on cultivated (i.e. tomato) plants.

Our response: Thank you for the suggestion. We read the suggested paper. Of course, we can understand the purpose of the study. However, the suggested study focused on olfactory response without field trial and verification of feeding damage decrease to plant. So, this paper was not included. However, we included a suggested statement and reference in our discussion. Please check L427-429, 515-518, and 674 in the revised manuscript.

13. Reviewer comment: L54: citations on prey and alternative prey availability need to be included (e.g., DOI 10.1007/s10526-015-9700-5. Doi: 10.1127/entomologia/2019/0824). Moreover, s the role of alternative and banker plants would need to be at least mentioned here as well (e.g., DOI: 10.1127/entomologia/2019/0625; doi: DOI: 10.1080/09670874.2012.659229)

Our response: Thank you for the suggestion. We added a sentence with suggested references. Please check L63-70, 600, 639, 642, and 644 in the revised manuscript.

Added sentence: Alternative food sources and use of banker plants can be used to help conserve natural enemies.

14. Reviewer comment: L56-58: confusing statements. Please rephrase

Our response: Thank you. Rephrase done. Please check L68-70 in revised manuscript.

Original sentence: To enhance settlement and persistence of natural enemies, escape of natural enemies should be minimized in the absence of food, and their aggregation should be maximized in the target area.

Revised sentence: Also, it is helpful to minimize emigration of natural enemies leaving the crop and to maximize natural enemy aggregation where pests are most abundant. 

15. Reviewer comment: L67: change “establishment of N. tenuis” to “establishment of this predatory mirid bug”

Our response: Thank you. In the English editing process, the original sentence was corrected. Please check L82-85 in the revised manuscript.

Original sentence: If this LED strategy can be applicable to N. tenuis in greenhouses, it could enhance the establishment of N. tenuis in target areas and help control pests such as B. tabaci.

Revised sentences: Given that this strategy [44] was successful in open agricultural fields, we hypothesized that the same use of LED light sources should work equally well in greenhouses. Our goal, therefore, was to assess the use of LED lights for enhancement of the impact of N. tenuis in greenhouses for control of pest (i.e., B. tabaci).

16. Reviewer comment: L68: change “help control pests such as B. tabaci” to “help pest (i.e., B. tabaci) control”

Our response: Thank you. In the English editing process, the original sentence was corrected with the suggestion. Please check L82-85 in the revised manuscript.

Original sentence: If this LED strategy can be applicable to N. tenuis in greenhouses, it could enhance the establishment of N. tenuis in target areas and help control pests such as B. tabaci.

Revised sentences: Given that this strategy [44] was successful in open agricultural fields, we hypothesized that the same use of LED light sources should work equally well in greenhouses. Our goal, therefore, was to assess the use of LED lights for enhancement of the impact of N. tenuis in greenhouses for control of pest (i.e., B. tabaci).

17. Reviewer comment: L68-69: In my opinion the sentence “In addition, ….. agent.” Is repetitive and not necessary here. I suggest deleting it.

Our response: Thank you. It was deleted. Please check L85 in the revised manuscript.

18. Reviewer comment: L74-75: as currently written seems that “enhancing spatial coherence … could successfully control B. tabaci”. I think a verb is needed between “successfully” and “control” (maybe improve, ameliorate, etc). Alternatively sentence modifications should be done to make sense to the statement.

Our response: Thank you. In the English editing process, the original sentence was corrected with the suggestion. Please check L90-95 in the revised manuscript.

Original sentence: Therefore, the objective of this study was to explore the proper wavelength of LED light to attract both B. tabaci and its predator, N. tenuis, and determine if enhancing spatial coherence of these two species using LED light could successfully control B. tabaci in a tomato greenhouse. 

Revised sentence: The specific objectives of this study, therefore, were to determine if selected wavelengths of LED light would enhance establishment of N. tenuis and thus improve biocontrol of B. tabaci in a tomato greenhouse. Additionally, the responses of B. tabaci to selected wavelengths of LED light were explored to determine their impact on the spatial congruence of the distributions of natural enemies and the target pest.

19. Reviewer comment: L77: change “Methods” to “methods”

Our response: Change done. Please check L97 in the revised manuscript.

20. Reviewer comment: L78: “Test insects” does not sound good to me. Please provide a better subtitle

Our response: We changed to “Insects sources and rearing”. Please check L98 in the revised manuscript.

21. Reviewer comment: L82. Spell out the whole genus name after “use.”

Our response: Thank you for the suggestion. However, according to your suggestion of #22, we added some words in front of N. tenuis. Thus, we did not change. Please check L103-104 in the revised manuscript.

22. Reviewer comment: L82: Please specify if the commercial N. tenuis specimens were used only to start the laboratory rearing. It is not clear in the current sentence text.

Our response: We mentioned the purpose of N. tenuis rearing after rearing methods. And, we recognized it could occur confusion to the reader. Thus, we added “For use in the incubator experiments (Exps. #2 and #3),” before “N. tenuis” and deleted the sentence “This N. tenuis adults were used for incubator experiments.”. Please check L103 and 110 in the revised manuscript.

23. Reviewer comment: L89: change “This” to “These” or start the sentence with some words like “Newly obtained N. tenuis adults…” or similar.

Our response: Thank you. However, we deleted that sentence according to the revision of reviewer comment #22. Please check L110 in the revised manuscript.

24. Reviewer comment: L90: see the comment of line 82

Our response: Thank you. We spelled out a genus name. Please check L110 in the revised manuscript.

25. Reviewer comment: L92: provide mean values ± S.D. or S.E. if possible

Our response: Thank you for your suggestion. Unfortunately, we do not have that data. Because the rearing place of B. tabaci was not where we managed it.

26. Reviewer comment: L100-101: provide specifications about position, size, orientation etc. of the transparent sticky trap inside Y-tube branches 

Our response: Thank you. We revised and added sentences. Please check L121-124 in the revised manuscript.

Original sentence: At each lighting zone, a transparent sticky trap was installed.

Revised sentences: At the end of each branch, a transparent sticky trap (40 mm dia) was installed. At the center of each of these transparent sticky traps, a hole (7 x 7 mm2; width x length) was drilled to reduce the interference of the trap for the light single. 

27. Reviewer comment: L102: how these LED light were applied (position, distance, lighted surface)? please specify

Our response: We added the sentences about your suggestion. Please check L124-126 in the revised manuscript.

Added sentence: The LED lights were installed in cups that inserted tightly into the Y-tube branches’ ends (45 mm ext. dia). The distance between the transparent sticky traps and the LED light source was 10 mm. 

28. Reviewer comment: L109: specify if 30 specimens were used for each replication (L106) (for a total of 150 specimens per tested wavelength) or in total. It is not completely clear.

Our response: Thank you for the suggestion. We changed “test” to “replication”. About 30 N. tenuis were used for each replication. Please check L135 in the revised manuscript.

29. Reviewer comment: L110-111: 50 B. tabaci specimens per tested wavelength in total? had you replications? Please provide more details about this relevant trial aspect

Our response: Thank you for the suggestion. We changed “test” to “replication”. About 50 B. tabaci were used for each replication. Please check L138 in the revised manuscript.

30. Reviewer comment: L117-118: confusing title, please rephrase

Our response: We changed the title by dividing two parts. Please check L146 and L204 in the revised manuscript.

Original title: Incubator experiment for verification of enhanced establishment rate

Revised title: 

Exp. #2: No-choice test of effect of light treatment on N. tenuis movement and predator efficiency 

Exp. #3: Choice test of effect of light treatment on N. tenuis movement and predator efficiency 

31. Reviewer comment: L178: the “Experimental design and sampling method” section is too long and sometimes redundant. Statement such as “Most UV-C from the sun is absorbed into the ozone layer.” could be deleted. General contents should focus on relevant aspects. I suggest rephrasing most of this section by writing concisely and linearly. Sentences of lines 202-206 should come before those of lines 199-202.

Our response: Thank you. We deleted that sentence. And moved those sentences. And, we rephrased our manuscript by English re-editing from “VanDriesche Scientific Editing”. Please check L218, 236-241 in the revised manuscript.

32. Reviewer comment: L228: you stated “June 19, the last release date of N. tenuis” but the last release was “on June 14”. Please verify and/or correct

Our response: Thank you. We revised that part of the sentence. Please check L257-259 in the revised manuscript.

Original part of sentence: the control value was calculated from June 19, the last release date of N. tenuis,

Revised part of sentence: the control value was calculated from June 19, the first observation date after the last release date (June 14) of N. tenuis,

33. Reviewer comment: L263-265: I suggest deleting percentage values from the statement “Five wavelengths attracted … > 405 nm (41.6%).” because it is repetitive. Indeed, reported results occur in table 2 and some in lines 261-262. (may be better: “385 nm > 365 nm > 445 ………”)

Our response: Thank you for the suggestion. We deleted. Please check L302-303 in the revised manuscript.

34. Reviewer comment: L268: I suggest adding a specification about the statement “except for wavelength at 445 nm”. As it is currently written, it could seem that 445 nm was the most attractive while it was the least attractive. Please modify or provide specifications

Our response: We added “, which showed the lowest attraction rate (37.6%)” after “445 nm”. Please check L307 in the revised manuscript.

35. Reviewer comment: L330-332: P values do not confirm that “not significantly different”. Verify and clarify

Our response: Actually, that is the results of RM ANOVA, not post hoc test (Tukey’s studentized range test). We presented the results of post hoc test in Fig 3. In Fig 3, there were four kinds of plots, and we just mentioned only three plots in that sentence.

36. Reviewer comment: L337-339: this should be moved to the discussion

Our response: Thank you for the suggestion. We moved that sentence to our discussion section with some revision. Please check L476-479 in the revised manuscript.

Original sentence: These results indicate that B. tabaci seems to have a different trend of population dynamics under 385 nm wavelength LED light depending on the presence or absence of natural enemies.

Revised sentences: In the first greenhouse trial (Exp. #4), we found that 385 nm wavelength LED lights seems to enhance the establishment rate of B. tabaci in absence of predator releases, however, this effect was reversed when both lighting and predators were employed together (Fig 3). 

37. Reviewer comment: L354-355: “seemed to help the control the activity of N. tenuis”. Provide correction

Our response: Revision done. We revised that sentence. Please check L387-388 in the revised manuscript.

Original sentence: In N. tenuis released plots, 385 nm wavelength LED light seemed to help the control the activity of N. tenuis for B. tabaci.

Revised sentence: In the N. tenuis release plots, use of 385 nm LED light seemed to enhance predation by N. tenuis on B. tabaci.

38. Reviewer comment: L357-358: these statements should be moved to the discussion

Our response: Thank you for the suggestion. We moved that sentence to our discussion section with some revision. Please check L479-483 in the revised manuscript.

Original sentence: It might be due to a later colonization of N. tenuis caused by an indeterminate reason such as temperature.

Revised sentences: LED lighting at a wavelength of 385 nm seemed to increase the control efficacy of the natural enemy (Exp. #4, Fig 5). However, the densities of predator in our first greenhouse trial (Exp. #4) was not affected by LED treatment. This lack of impact on the predator may have been due to the later colonization of N. tenuis of our plots due to low temperature or low whitefly density early in the trial. 

39. Reviewer comment: L390-391: this statement should be moved to the discussion

Our response: This statement is already mentioned in our discussion. Thus, we deleted that sentence. Please check L411 in the revised manuscript.

40. Reviewer comment: L411-412: spell out the genus after the point “preference. N.”

Our response: Thank you. Revised done. Please check L440 in the revised manuscript.

41. Reviewer comment: L452-454: confusing sentence. Rephrase

Our response: Thank you. We revised our sentence. Please check L494-495 in the revised manuscript.

Original sentence: In the plot with 385 nm wavelength LED light after the release of N. tenuis and B. tabaci, the 385 nm wavelength LED light appeared to enhance the establishment of N. tenuis.

Revised sentence: In plots with supplemental 385 nm wavelength LED light and the release of both N. tenuis and B. tabaci, the LED light appeared to enhance the establishment of N. tenuis.

Reviewer’s comments and our responses (Reviewer #3)

1. Reviewer comment: The manuscript “Increase of control efficacy of Nesidiocoris tenuis (Hemiptera: Miridae) in the greenhouse by enhancing its establishment using UV-LED light”, study the attraction and effect of Led Light on Nesidiocoris tenuis and the herbivore Bemisia tabaci, under laboratory and greenhouse conditions. In general, the objective of the work seems interesting to me. However, I have doubts about recommending publishing this paper in PlosOne.

In general, I found some problems with this manuscript in terms of title, the purpose of study, experiments, and results from interpretation. Thus, I really have some concerns about this work, which could be a problem to publish in this journal.

At least for this reviewer, the ms needs a review of the English language. Furthermore, some wordings are challenging to understand ("spatial coherence", “settlement”, "spatial separation", etc…).

Our response: Thank you for the comments. We sincerely appreciate your suggestions. We tried to supplement our manuscript and respond sincerely to your comments. According to reviewers’ suggestion, our manuscript was professionally re-edited for English by "VanDriesche Scientific Editing".

Title

2. Reviewer comment: The title is tough to understand, at least for this reviewer. I suggest to try with something like this: “LED lights enhance the establishment and biological control efficacy of N. tenuis”. 

Our response: Thank you for the suggestion. We changed our title after referring to your opinion. Please check L1-2 in the revised manuscript.

Original title: Increase of control efficacy of Nesidiocoris tenuis (Hemiptera: Miridae) in the greenhouse by enhancing its establishment using UV-LED light

Changed title: UV-LED Lights Enhance the Establishment and Biological Control Efficacy of Nesidiocoris tenuis (Reuter) (Hemiptera: Miridae) in the Greenhouse

Introduction

3. Reviewer comment: Authors introduce both insects well; however, at least for this reviewer, the introduction section should have a better hypothesis of the work and made the objectives clear.

Our response: Thank you. We tried to improve our introduction according to reviewers’ suggestion. Please check L86-95 in the revised manuscript.

4. Reviewer comment: Line 40-42: This paragraph should move below after to introduce N. tenuis.

Our response: Thank you. We moved those sentences to the end of the paragraph with revision. Please check L60-62 in the revised manuscript.

Original sentences: Generalist natural enemies have alternative food sources besides their target pests. Thus, they have higher adaptability to sustain in the target area than specialist natural enemies.

Revised sentences: Because generalist natural enemies can subsist on alternative food sources when the target pest is rare, they can be more adaptable to changing circumstances in the crop than specialist natural enemies.

5. Reviewer comment: Line 50-52: There are also alternative food sources as sugars that reduce this damage in tomato plants, authors should be a reference to these works...

Our response: Thank you for the suggestion. We found and read the paper (doi.org/10.1111/jen.12151). And, we revised our sentence with reference. Please check L60 and 631 in the revised manuscript.

Original sentence: Typical studies include reducing its damage to plants using an endophytic strain Fusarium solani and having a proper release density for N. tenuis.

Revised sentence: Related studies have been conducted to reduce the mirid’s risk to crop plants, including studies of plant resistance to N. tenuis mediated by endophytic strain of Fusarium solani K, application of sugar as a complementary alternative food, and optimizing the release density for N. tenuis. 

6. Reviewer comment: Line 59: There are also alternative food sources as sugars that improve the establishment of N. tenuis in tomato plants; authors should be a reference to these works...

Our response: Thank you for the suggestion. We revised our sentence. Please check L70-74 in the revised manuscript.

Original sentence: Supplementary food spray using nectar plant and release of natural enemies to the transplant tray a few days before planting can be attempted.

Revised sentence: Manipulations tested to improve the spatial association of natural enemies with the pests they attack include the release of the natural enemies onto the transplant trays (when plants are most concentrated) a few days before planting and application of supplementary foods such as plant nectar or sugar to areas with high pest infestations.

7. Reviewer comment: Line 66: Authors should make the hypothesis of the study more clear

Our response: Thank you. We supplemented that part with the help of the reviewer and English editing company. Please check L82-95 in the revised manuscript.

Original: If this LED strategy can be applicable to N. tenuis in greenhouses, it could enhance the establishment of N. tenuis in target areas and help control pests such as B. tabaci. In addition, it can increase the value of N. tenuis as a biological control agent. However, we have to consider the reactions of pest species to selected wavelength. If a selected LED light shows asymmetrical response between N. tenuis and its prey, it could result in spatial separation between the predator and the prey. Therefore, the objective of this study was to explore the proper wavelength of LED light to attract both B. tabaci and its predator, N. tenuis, and determine if enhancing spatial coherence of these two species using LED light could successfully control B. tabaci in a tomato greenhouse.

Revision: Given that this strategy [44] was successful in open agricultural fields, we hypothesized that the same use of LED light sources should work equally well in greenhouses. Our goal, therefore, was to assess the use of LED lights for enhancement of the impact of N. tenuis in greenhouses for control of pest (i.e., B. tabaci). However, the reaction of key pest species to selected wavelength also have to be considered. If a selected LED light is asymmetrically attractive between N. tenuis and its prey, the spatial separation between the predator and the prey, rather than being reduced, might increase. Therefore, we hypothesized that the best strategy would be to select LED light wavelengths that have high attractiveness for both the predator and prey. The specific objectives of this study, therefore, were to determine if selected wavelengths of LED light would enhance establishment of N. tenuis and thus improve biocontrol of B. tabaci in a tomato greenhouse. Additionally, the responses of B. tabaci to selected wavelengths of LED light were explored to determine their impact on the spatial congruence of the distributions of natural enemies and the target pest.

8. Reviewer comment: Line 72: Why the authors want to attract B. tabaci to tomato plants?? Could this attraction be a detriment for N. tenuis biocontrol?

Our response: Thank you for the nice question. However, unlike pheromone that can attract specific species, it is tough to induce attraction response to one species using a certain wavelength of light. There were many reports about the positive response for UV or near-UV for pests and natural enemies. Some wavelengths of those ranges were finally chosen for enhancing the establishment of natural enemies [1-3]. In these results, applying a certain wavelength to attract specific species of insects might be hard. Thus, it would be better to consider the spatial congruence of both pests and natural enemies for enhancing the efficacy of biological control. Please check L82-95 in the revised manuscript. It is probably helpful.

In our greenhouse experiment, we closed side windows of greenhouses when LED lights were turned on, and the windows were opened after sunset to reduce the external inflow of B. tabaci by lighting.

References

1. Johansen NS, Vänninen I, Pinto DM, Nissinen AI, Shipp L. In the light of new greenhouse technologies: 2. Direct effects of artificial lighting on arthropods and integrated pest management in greenhouse crops. Ann Appl Biol. 2011; 159(1): 1-27.

2. Ogino T, Uehara T, Muraji M, Yamaguchi T, Ichihashi T, Suzuki T, et al. Violet LED light enhances the recruitment of a thrip predator in open fields. Sci Rep. 2016; 6: 32302.

3. Uehara T, Ogino T, Nakano A, Tezuka T, Yamaguchi T, Kainoh Y, et al. Violet light is the most effective wavelength for recruiting the predatory bug Nesidiocoris tenuis. BioControl. 2019; 64(2): 139-147.

M&M

9. Reviewer comment: Line 81: “25ºC” needs (± SE). Here and throughout the manuscript. 

Our response: Before we used it, the incubator’s temperature was set at 25 ºC for a long time. And, in our opinion, the storage time was not too long. Thus, we didn’t check the variation of temperature for this method.

10. Reviewer comment: Line 82: “at 25ºC for ca. 1h before use”, at least for this reviewer, this is a little bit misleading if these N. tenuis are for rearing.

Our response: Thank you for the comment. Actually, that condition was set for storage of N. tenuis before Y-tube test. We changed “ca.” to “about”. And, we added, “For use in the incubator experiments (Exps. #2 and #3),” to the beginning of the next sentence. Please check L102-103 in the revised manuscript.

11. Reviewer comment: Line 88: Why the authors use eggs of C. cautella to evaluate the predation rate instead of other eggs source (e.g. Tuta absoluta, Ephestia kuehniella)?

Our response: In our situation, using eggs of C. cautella was the best option. Because we can easily buy the eggs when we want from Osang Kinsect System. And, Nesidiocoris tenuis is a generalist predator, and it can use most of Pyralid moth eggs as prey. Moreover, Tuta absoluta is not live in Korea. 

12. Reviewer comment: Line 89: Which N. tenuis? How old are N. tenuis used in ALL the experiments? Are males or females?

Our response: We deleted that sentence according to the suggestion of Reviewer 2. Instead, we added, “For use in the incubator experiments (Exps. #2 and #3),” at the beginning of the rearing part. For the experiments, we collected N. tenuis by random. Please check L103 and 110 in the revised manuscript.

13. Reviewer comment: Line 96: Although the Y-tube results seem that the response of B. tabaci and N. tenuis are reliable enough. I think that the methodology of this experiment needs to be more explicit. How the authors separated the “lighting zone”? Is there any work that used this methodology for light attraction?

Our response: In our Y-tube, there were three branches. It was made by 3D printing of plastic, and the outer cover of the Y-tube was painted in black. Chiel et al. (2006) used a similar method. We newly cited Chiel et al. (2006). And we revised and added a sentence for this. Please check L118-120, 130-132 and 667 in the revised manuscript.

Reference

Chiel E, Messika Y, Steinberg S, Antignus Y. The effect of UV-absorbing plastic sheet on the attraction and host location ability of three parasitoids: Aphidius colemani, Diglyphus isaea and Eretmocerus mundus. BioControl. 2006; 51: 65–78.

Original sentence: During the test, each branch of the Y-tube was capped to block penetration of light from the outside.

Revised sentences: During a test, each branch of the Y-tube was capped to prevent penetration of light from the outside into the test arena. The body of the Y-tube made of plastic (3D printing) that did not permit light to pass.

14. Reviewer comment: Line 100: “branches” means “arms”?

Our response: Yes.

15. Reviewer comment: Line 108: “About 30 N. tenuis”?. The authors need to be more concise.

Our response: For every replication of our Y-tube test, we collected the insects using an electric aspirator by counting the number of insects. Some insects were escaped in the process of input to Y-tube, and sometimes, insects were collected more than that we counted. It was tough to count in the bottle of aspirator because it was always moving. Of course, we can make them knock out in a moment by freezing. However, we thought that it would be some stress on them. Moreover, we just counted again after the experiment. So, there was some variance in the insect number. That is why we used “about” or “approximately” in lines 108 and 110 of the original manuscript.

16. Reviewer comment: Line 110: “Approximately 50 B. tabaci”?. The authors need to be more concise.

Our response: Please check our answer for reviewer comment #15.

17. Reviewer comment: Line 117: this section evaluates the predation rate. It is okay. However, how the authors evaluate the establishment?

Our response: As we mentioned in the last paragraph of that section, the incubator was not sealed. There were many holes for ventilation and environmental control. Thus, the mirid bugs can escape using that hole. In our results, 385 nm wavelength treatment can make more N. tenuis stay in the incubator. Thus, we assume that 385 nm wavelength LED light can increase the establishment of N. tenuis in a certain area. However, we partly agreed about your comments. Thus, we gave some revision for that part by deleting “establishment”. Please check L150-155 in the revised manuscript.

Original sentence: predation amount and establishment rate of N. tenuis were compared among different light conditions in the incubator.

Revised sentences: In this experiment, we measured the rate of predation by the released mirids on eggs of C. cautella as a “predation index”. The incubator was not a fully enclosed space. Rather, it had some holes in the inner side wall for environmental controls, and there was a vessel of water behind one wall. Some N. tenuis left the experimental arena and drowned in this water. Thus, we also counted the remaining number of mirids in an incubation chamber at the end of each assay.

18. Reviewer comment: Line 152: Which is the second experiment?

Our response: We divided that section. Exp. #3 is the second experiment. Please check L146 and 204 in the revised manuscript.

19. Reviewer comment: Line 198: N. tenuis population includes nymphs and adults? Also, for the predation experiment?

Our response: In the greenhouse test, N. tenuis population includes nymphs and adults. However, in the incubator test, we used the only adult. Because we want to check how many N. tenuis can escape in an incubator. The adults have wings, so we thought that adult N. tenuis is more suitable for our purpose.

20. Reviewer comment: Line 199: all the tomato plant leaves were counted??

Our response: Yes, we checked all leaves, stems, and trunks.

21. Reviewer comment: Line 203-204: “7 to 20 N. tenuis were release”? The number of N. tenuis released is essential for results interpretation.

Our response: We represented all the release number and date in Table 1. Please check L246 in the revised manuscript.

22. Reviewer comment: Line 204-205: same for the number of B. tabaci

Our response: We represented all release the number and date in Table 1. Please check L246 in the revised manuscript.

23. Reviewer comment: Line 221: Calvo et al. 2012 showed that 0.25-0.5 N. tenuis per plant was a good release range.

Our response: Yes, right. However, that release range is for pre-plant release strategy. Actually, we did not apply that strategy in our experiments.

Results

24. Reviewer comment: Line 258: Removed this sentence

Our response: Thank you for the suggestion. We removed it. Please check L297 in the revised manuscript.

25. Reviewer comment: Table 2: It is known that results from Y-tube should be shown in vertical bars figures. For this reason, at least for this reviewer, results from the Y-tube will be better represented by figures (e.g. left side responders to different LED light treatments, right side responders to control treatment, and outside the figure but next to it, the number of individuals who did not respond should be included).

Our response: Thank you for the suggestion. We think you might want to revise Table 2 to the figure (extracted from Naselli et al. 2016) below.

In the results of Naselli et al. (2016), almost all insects could make their choice. Thus, they only compared two treatments. However, in our results, the treatments of different wavelengths significantly affected to no response rate. Thus, we think that no response is also an important factor. And, we think representing in the Table is better for comparing among the different wavelength treatments.

Reference

Naselli M, Zappala L, Gugliuzzo A, Garzia G T, Biondi A, Rapisarda C, Cincotta F, Condurso C, Verzera A, Siscaro G. Olfactory response of the zoophytophagous mirid Nesidiocoris tenuis to tomato and alternative host plants. Arthropod-Plant Interactions, 2016; 11(2): 121-131.

Greenhouse experiments

26. Reviewer comment: Why did the authors not use a GLMM to analyze the greenhouse data?? This model can include factors as the date and interaction date*treatment…

Our response: Thank you for the comments. For the Generalized linear mixed model (GLMM) in SAS, PROC MIXED should be used. And there are two options for RM ANOVA in SAS, PROC GLM and MIXED. They have pros and cons. 

 Firstly, we analyzed our greenhouse data by using PROC GLM. And we analyzed it again with PROC MIXED. However, the statistical values (i.e., F-value, df, P-value) were the same. So we just wrote the PROC GLM that we analyzed first in our manuscript. Moreover, PROC GLM also include factors as the date and interaction date*treatment, and we already mentioned the results of date and date*treatment in our results section. Please check L352-355, 372-375, and 402-406 in the revised manuscript.

27. Reviewer comment: Line 354: “In N. tenuis released plots, 385 nm wavelength LED light seemed to help the control the activity of N. tenuis for B. tabaci”. However, when you showed the figure, there are no differences between treatments in B. tabaci populations…It seems that LED lights did not affect B. tabaci control by N. tenuis.

Our response: Thank you for the comments. Actually, in our Exp. #4 (revised version of greenhouse exp. 1), we can’t find a significant difference in N. tenuis density. Thus, we suggested the control value (Fig 5). We think it can be supplementary data for Exp. #4. And we discussed this in our discussion section (L476-489 in the revised manuscript). We thought that this result was caused by two reasons: 1. The Low density of B. tabaci and 2. Environmental condition. The initial density of B. tabaci in Exp. #4 was very low. Thus N. tenuis cannot make their population increase. Also, the early temperature of Exp. #4 was lower than Exp. #5. Low temperature affected to population development of both B. tabaci and N. tenuis. Thus, we released N. tenuis and B. tabaci three times in Exp. #4. From June mid, both insect densities were increased in all plots. We could get significant results in the density of B. tabaci. However, it was too late to get a significant result for N. tenuis. Thus, we concentrated on N. tenuis density and its control efficacy in Exp. #5. That is why we organized different plots between Exp. #4 and #5.

28. Reviewer comment: Line 368: At least for this reviewer, it is hard to understand how the percentage of control value of B. tabaci + N. tenuis + LED are higher in all the sample dates if the B. tabaci levels are not significantly different.

Our response: Thank you for the question. When we calculated the control value, we used B. tabaci only plot as a control, and another two plots (B. tabaci + N. tenuis / B. tabaci + N. tenuis + LED) were treatments. Mean densities were used for control value without regarding statistical significance. Thus, we can find some differences in our control value. We suggested control value as a supported data for N. tenuis and necessity of Exp. #5. Because we cannot get a significant result of N. tenuis in greenhouse Exp. #4. We explained the reason in the answer of the above question and our discussion (L476-489 in the revised manuscript). 

29. Reviewer comment: Line 371: The experiment 2 showed that LED lights' effect on B. tabaci populations reductions correlated with N. tenuis populations increases. The authors should justify better the results of the first experiment throughout the discussion.

Our response: Thank you for the suggestion. Please check L476-489 in the revised manuscript.

Added sentences: In the first greenhouse trial (Exp. #4), we found that 385 nm wavelength LED lights seems to enhance the establishment rate of B. tabaci in absence of predator releases, however, this effect was reversed when both lighting and predators were employed together (Fig 3). LED lighting at a wavelength of 385 nm seemed to increase the control efficacy of the natural enemy (Exp. #4, Fig 5). However, the densities of predator in our first greenhouse trial (Exp. #4) was not affected by LED treatment. This lack of impact on the predator may have been due to the later colonization of N. tenuis of our plots due to low temperature or low whitefly density early in the trial. 

Discussion

30. Reviewer comment: In my consideration, the discussion should be rewritten again. Authors need to be more consistent with their results and then discuss better with other works. There are a large number of sentences that need references.

Our response: Thank you for the comments. We supplemented our discussion with re-editing in English. And we added some references. Please check L427-430, 437-438, 451, 468, 476-489, 496, 497, 502, 504, and 518 in the revised manuscript.

31. Reviewer comment: Line 421 -422 and Line 433-434 are almost the same sentences. In my opinion, I removed the first one because this sentence here did not match the flow of the discussion.

Our response: Thank you for the suggestion. However, supplementary light is not always used by combining red and blue light. We intend in the first sentence (Line 421-422) to use each wavelength separately, and we think that deleting the first sentence will not support the following sentences. The best option we have in mind is to change the latter part of the second sentence (Line 433-434 in the original manuscript). Please check L458-459 in the revised manuscript.

Original sentence: Supplementary light for promoting plant growth is often used by combining red and blue light

Revised sentence: Supplementary light for promoting plant growth often based on a combination of different wavelengths of light.

32. Reviewer comment: Line 439 and line 443: “supported results of the greenhouse experiment” (experiment 1 or 2?)

Our response: Thank you. As a result, that is proper content for Exp. #5 (revised version of experiment 2), but we intended and wrote both Exp. #4 and #5 because we obtained similar results from the whitefly in Exp. #4 (revised version of experiment 1). 

33. Reviewer comment: Line 450: Figure 8. From where are these temperature and humidity data? Which greenhouse? If authors represent each day's abiotic conditions, they should put the temperature, and RH mean and SE.

Our response: We installed the data logger in each greenhouse. Thus, we added “of each greenhouse” in M&M. Our data is mean data between observation date. And we revised our figure. Please check L243, 491 in the revised manuscript and newly updated Fig 8.

34. Reviewer comment: Line 468-469: Experiment 1 or 2?

Our response: It means the whole of our study, not just for the greenhouse experiment.

---

## [Decision Letter · Decision Letter 1]

16 Dec 2020

PONE-D-20-30203R1

UV-LED Lights Enhance the Establishment and Biological Control Efficacy of Nesidiocoris tenuis (Reuter) (Hemiptera: Miridae) in the Greenhouse

PLOS ONE

Dear Dr. Lee,

Thank you for submitting your manuscript to PLOS ONE. After careful consideration, we feel that it has merit but does not fully meet PLOS ONE’s publication criteria as it currently stands. Therefore, we invite you to submit a revised version of the manuscript that addresses the points raised during the review process.

Your manuscript has been re-reviewed by two referees. They both provided very important criticisms, especially on the suitability of the greenhouse experiment. I concur with them on the fact that such portion of the manuscript should be deleted and the whole document revised according to this major change.

We look forward to receiving your revised manuscript.

Kind regards,

Antonio Biondi, Ph.D.

Academic Editor

PLOS ONE

Reviewers' comments:

Reviewer's Responses to Questions

**Comments to the Author**

1. If the authors have adequately addressed your comments raised in a previous round of review and you feel that this manuscript is now acceptable for publication, you may indicate that here to bypass the “Comments to the Author” section, enter your conflict of interest statement in the “Confidential to Editor” section, and submit your "Accept" recommendation.

Reviewer #2: (No Response)

Reviewer #3: All comments have been addressed

2. Is the manuscript technically sound, and do the data support the conclusions?

Reviewer #2: Partly

Reviewer #3: Partly

3. Has the statistical analysis been performed appropriately and rigorously? 

Reviewer #2: N/A

Reviewer #3: N/A

4. Have the authors made all data underlying the findings in their manuscript fully available?

Reviewer #2: No

Reviewer #3: No

5. Is the manuscript presented in an intelligible fashion and written in standard English?

Reviewer #2: Yes

Reviewer #3: No

6. Review Comments to the Author

Reviewer #2: Authors re-submitted a new version of the manuscript where they ameliorated some minor concerns. The current version appears properly written and referenced, even if some changes are still necessary (see specific comments).

However, I renew my strong doubts about the methods employed for conducting the greenhouse trials (occurrence of the pest before starting trials, varying number of insect pest released in different times, release of very few predators) and consequently I confirm my first recommendation to delete (from all relevant sections, included the title) the greenhouse part of this manuscript. Indeed, in my opinion, all the above reasons do not allow to have a sound experimental setup.

Specific comments:

L21: add “its” before “co-occurrence”

L21-22: add “programs” before “in greenhouses”

L24: change “crop” to “crops”

L67: I suggest to delete “and use” and to say directly “Alternative food sources and banker plants can be used…”

L72: add “the” before “application”

L85: I suggest to use “for pest (i.e., B. tabaci) control”

L122 and 125: the use of the abbreviation in “40 mm “dia”” and “45 mm “ext. dia”” does not sound good to me, please check the journal guidelines and make changes if necessary

L131-132: a verb seems to be missing in this sentence, please correct it

L218: may be “considered” despite of “consider”?

L220: may be “consider” and “tested” despite of “considered” and “test”?

L227: change contained” to “consisted of”

L246 - Table 1: Were the releases of “B. tabaci + N. tenuis +LED” the same of “B. tabaci + N. tenuis”? Please, make the Table more intelligible and clarify/specify the aforementioned doubt

L276: correct “desired”

L287: change “Mean” to “mean”

Reviewer #3: Although the authors have improved the paper compared to the first submission, at least for me, they have not made enough adjustments to have the manuscript ready for acceptance.

My primary concern is the part of the greenhouse experiments. At least for this reviewer, the methodology of greenhouse experiments is misleading and is not strong enough to be published in this journal (i.e., the numbers of the whiteflies and N. tenuis used in the experiments). I read the reviewer 2 comments, and this reviewer recommended removing the greenhouse experiments (which I partly agree with), however authors did not remove it. At least for this reviewer, authors should do the experiment again under greenhouse conditions or try to remove it from the study; however, this reviewer is not sure if the study will be too short. Furthermore, the answer to points 15 and 16 (of the reviewer 3) made by authors is questionable concerning this topic.

15. Reviewer comment: Line 108: “About 30 N. tenuis”?. The authors need to be

more concise.

16. Reviewer comment: Line 110: “Approximately 50 B. tabaci”?. The authors need to be more concise.

Our response: For every replication of our Y-tube test, we collected the insects using

an electric aspirator by counting the number of insects. Some insects were escaped

in the process of input to Y-tube, and sometimes, insects were collected more than

that we counted. It was tough to count in the bottle of aspirator because it was always

moving. Of course, we can make them knock out in a moment by freezing. However,

we thought that it would be some stress on them. Moreover, we just counted again

after the experiment. So, there was some variance in the insect number. That is why

we used “about” or “approximately” in lines 108 and 110 of the original manuscript.

There are many ways to know the total number of insects used in the experiments. At least for this reviewer, the numbers of insects used for the experiments are very important and help make the study more concise for a potential publication in a scientific journal.

However, I consider that I mistake by correcting when I suggest representing the Y-tube results (Table 2 and 3) in a figure with vertical bars. I want to say horizontal bars (similarly to Naselli's figure that the authors used to answer my suggestion), with the non-responders individuals between brackets in the corresponding bars.

Minor comments

- Change “Exp. #1” to “Experiment 1” throughout all the manuscript, experiments (2, 3, and 4), including the subheadings.

- Change “branches” by “arms” about the y-tube.

- Why are the ±SE of the abiotic conditions (Temperature and relative humidity) too high?

7. PLOS authors have the option to publish the peer review history of their article (what does this mean?). If published, this will include your full peer review and any attached files.

Reviewer #2: No

Reviewer #3: No

---

## [Author Response · Author response to Decision Letter 1]

23 Dec 2020

Dear Editor

Thank you for giving us another chance and the helpful comments on our manuscript. We have faithfully revised our manuscript by referring to the reviewer's opinion. After careful consideration, we decided to delete the part of the greenhouse experiment in our manuscript. Along the way, we revised the title to "UV-LED Lights Enhance the Establishment and Biological Control Efficacy of Nesidiocoris tenuis (Reuter) (Hemiptera: Miridae)". And, we gave the new number to our figures and table. Moreover, according to the suggestion of Reviewer 3, we changed tables (about the Y-tube test) to figures. 

Revised parts for removing the greenhouse experiments, reviewer 2, and 3 were marked in red, green, and orange color in the text, respectively.

Revised part by deleting the greenhouse experiments

1. Title: revised.

Previous title: UV-LED Lights Enhance the Establishment and Biological Control Efficacy of Nesidiocoris tenuis (Reuter) (Hemiptera: Miridae) in the Greenhouse

Revised title: UV-LED Lights Enhance the Establishment and Biological Control Efficacy of Nesidiocoris tenuis (Reuter) (Hemiptera: Miridae)

Abstract

2. L23 in the revised manuscript: “in laboratory condition” added after “establishment of N. tenuis”

3. L28 in the revised manuscript: deleted “and lights of ~ in greenhouses” after “the attraction of N. tenuis to 385 nm wavelength”

4. L28–31 in the revised manuscript: revised.

Original sentence: When LED light at a wavelength of 385 nm was used in a greenhouse for 6 hours starting at sunset every night, it significantly enhanced the establishment rate of N. tenuis and the level of control of B. tabaci achieved in comparison to treatment without LED light.

Revised sentence: When LED light at a wavelength of 385 nm was used in a growth chamber for 6 hours out of 24 hours, it significantly increased the remaining number of N. tenuis in growth chamber and level of predation compared to treatment with white LED light or without LED light.

5. L31-34 in the revised manuscript: revised.

Original sentence: In conclusion, UV-LED light at a wavelength of 385 nm attracts both B. tabaci and N. tenuis, resulting in early establishment of this predatory mirid bug, better spatial congruence of both species, and better control of the whitefly.

Revised sentences: In conclusion, UV-LED light at a wavelength of 385 nm attracts both B. tabaci and N. tenuis. Thus, it would be used for enhancing early establishment of this mirid bug, better spatial congruence of both mirid bug and whitefly, and better control of the whitefly.

Introduction

6. L82-84 in the revised manuscript: revised.

Original sentence: Our goal, therefore, was to assess the use of LED lights for enhancement of the impact of N. tenuis in greenhouses for control of pest (i.e., B. tabaci).

Revised sentence: Our goal, therefore, was to evaluate the use of LED lights for enhancement of the impact of N. tenuis in the laboratory condition for pest (i.e., B. tabaci) control. 

7. L91 in the revised manuscript: deleted “and thus improve biocontrol of B. tabaci in a tomato greenhouse” after “would enhance establishment of N. tenuis”

Materials and methods

8. L148-151 in the revised manuscript: moved from the greenhouse experiment section.

9. L215, 224 in the revised manuscript: the greenhouse part deleted (L214-277 and 287-293 in the original manuscript were deleted).

Results

10. L238-239 in the revised manuscript: revised.

Original sentence: Thus, the 385 nm wavelength was selected for the greenhouse experiment.

Revised sentence: Thus, the 385 nm wavelength was selected for the incubator assays.

11. L281 in the revised manuscript: the greenhouse part deleted (L350-424 in the original manuscript were deleted).

Discussion

12. L299 in the revised manuscript: added “and accomplishing their spatial congruence” after “for further test”.

13. L322 in the revised manuscript: deleted “and this finding supported the results of our greenhouse experiments” after “incubator assays.”.

14. L327-328 in the revised manuscript: revised.

Original sentence: These results also support the results of our greenhouse experiments.

Revised sentence: Thus, if this LED light is applied in the greenhouse, it may increase the retention time of N. tenuis resulting in enhanced establishment of this predator.

15. L332-333 in the revised manuscript: Related paragraph and figure legend about the greenhouse experiments were deleted (L476-497 in the original manuscript were deleted).

16. L337-339 in the revised manuscript: revised.

Original sentences: When the abundance of natural enemies is low due to the use of pesticides or a naturally low number of natural enemies in the crop, lighting condition may accelerate population increase of the target pests (Fig 3). In this study, we closed the greenhouse’s vents during the supplemental lighting period to reduce the attraction of whiteflies into the greenhouse.

Revised sentences: Thus, the greenhouse's vents should be closed during the supplemental lighting period to reduce the attraction of whiteflies into the greenhouse, and regular monitoring of both natural enemy and pest densities would be needed.

17. L342-345 in the revised manuscript: revised.

Original sentences: Results of this study clearly indicate that 385 nm wavelength LED light can improve the efficacy of N. tenuis in greenhouses for controlling B. tabaci. The 385 nm wavelength LED light appeared to attract N. tenuis and enhance early population build-up of N. tenuis, resulting in better control of B. tabaci.

Revised sentences: Results of this study clearly indicate that 385 nm wavelength LED light can improve the efficacy of N. tenuis in released area. Also, the 385 nm wavelength LED light could help achieve the spatial congruence of both N. tenuis and B. tabaci in the target area because this LED light can attract both species. 

18. L353-355 in the revised manuscript: revised.

Original sentence: Further studies are needed to better understand how best to use LED lights, such as optimal lighting time and wattage of lights and assessments of the applicability of such light supplementation in other greenhouse crops.

Revised sentence: Further studies are needed to better understand how best to use LED lights, such as optimal lighting time and wattage of lights and assessments of the applicability of such light supplementation in the greenhouse.

Reviewer’s comments and our responses (Reviewer #2)

1. Reviewer comment: Authors re-submitted a new version of the manuscript where they ameliorated some minor concerns. The current version appears properly written and referenced, even if some changes are still necessary (see specific comments). However, I renew my strong doubts about the methods employed for conducting the greenhouse trials (occurrence of the pest before starting trials, varying number of insect pest released in different times, release of very few predators) and consequently I confirm my first recommendation to delete (from all relevant sections, included the title) the greenhouse part of this manuscript. Indeed, in my opinion, all the above reasons do not allow to have a sound experimental setup.

Our response: We deleted the part of the greenhouse experiment in our manuscript. Please check the part "Revised part by deleting the greenhouse experiments" before this. And, we revised our manuscript according to your specific comments. 

2. Reviewer comment: L21: add “its” before “co-occurrence”

Our response: Yes. Please check L21 in the revised manuscript.

3. Reviewer comment: L21-22: add “programs” before “in greenhouses”

Our response: Yes. Please check L22 in the revised manuscript.

4. Reviewer comment: L24: change “crop” to “crops”

Our response: Yes. Please check L24 in the revised manuscript.

5. Reviewer comment: L67: I suggest to delete “and use” and to say directly “Alternative food sources and banker plants can be used…”

Our response: Yes. Please check L66 in the revised manuscript.

6. Reviewer comment: L72: add “the” before “application”

Our response: Yes. Please check L71 in the revised manuscript.

7. Reviewer comment: L85: I suggest to use “for pest (i.e., B. tabaci) control”

Our response: Yes. Please check L84 in the revised manuscript.

8. Reviewer comment: L122 and 125: the use of the abbreviation in “40 mm “dia”” and “45 mm “ext. dia”” does not sound good to me, please check the journal guidelines and make changes if necessary

Our response: We revised “dia” to “diameter” and “ext. dia” to “external diameter”. Please check L120 and 123 in the revised manuscript.

9. Reviewer comment: L131-132: a verb seems to be missing in this sentence, please correct it

Our response: We added the verb “was”. Please check L129 in the revised manuscript.

10. Reviewer comment: L218: may be “considered” despite of “consider”?

Our response: That line was deleted by removing the greenhouse part in the manuscript.

11. Reviewer comment: L220: may be “consider” and “tested” despite of “considered” and “test”?

Our response: That line was deleted by removing the greenhouse part in the manuscript.

12. Reviewer comment: L227: change contained” to “consisted of”

Our response: That line was deleted by removing the greenhouse part in the manuscript.

13. Reviewer comment: L246 - Table 1: Were the releases of “B. tabaci + N. tenuis +LED” the same of “B. tabaci + N. tenuis”? Please, make the Table more intelligible and clarify/specify the aforementioned doubt

Our response: That part was deleted by removing the greenhouse part in the manuscript.

14. Reviewer comment: L276: correct “desired”

Our response: That was deleted by removing the greenhouse part in the manuscript.

15. Reviewer comment: L287: change “Mean” to “mean”

Our response: That was deleted by removing the greenhouse part in the manuscript.

Reviewer’s comments and our responses (Reviewer #3)

1. Reviewer comment: Although the authors have improved the paper compared to the first submission, at least for me, they have not made enough adjustments to have the manuscript ready for acceptance.

My primary concern is the part of the greenhouse experiments. At least for this reviewer, the methodology of greenhouse experiments is misleading and is not strong enough to be published in this journal (i.e., the numbers of the whiteflies and N. tenuis used in the experiments). I read the reviewer 2 comments, and this reviewer recommended removing the greenhouse experiments (which I partly agree with), however authors did not remove it. At least for this reviewer, authors should do the experiment again under greenhouse conditions or try to remove it from the study; however, this reviewer is not sure if the study will be too short. 

Furthermore, the answer to points 15 and 16 (of the reviewer 3) made by authors is questionable concerning this topic.

15. Reviewer comment: Line 108: “About 30 N. tenuis”?. The authors need to be

more concise.

16. Reviewer comment: Line 110: “Approximately 50 B. tabaci”?. The authors need to be more concise.

Our response: For every replication of our Y-tube test, we collected the insects using

an electric aspirator by counting the number of insects. Some insects were escaped

in the process of input to Y-tube, and sometimes, insects were collected more than

that we counted. It was tough to count in the bottle of aspirator because it was always

moving. Of course, we can make them knock out in a moment by freezing. However,

we thought that it would be some stress on them. Moreover, we just counted again

after the experiment. So, there was some variance in the insect number. That is why

we used “about” or “approximately” in lines 108 and 110 of the original manuscript.

There are many ways to know the total number of insects used in the experiments. At least for this reviewer, the numbers of insects used for the experiments are very important and help make the study more concise for a potential publication in a scientific journal.

However, I consider that I mistake by correcting when I suggest representing the Y-tube results (Table 2 and 3) in a figure with vertical bars. I want to say horizontal bars (similarly to Naselli's figure that the authors used to answer my suggestion), with the non-responders individuals between brackets in the corresponding bars.

Our response: Thank you for the comments. We sincerely appreciate your suggestions. After careful consideration, we decided to delete the part of the greenhouse experiment in our manuscript. Moreover, we tried not to diminish the manuscript much. Please check the part "Revised part by deleting the greenhouse experiments" before the response to Reviewer #2. 

Following your suggestion, we changed our tables for the Y-tube test to figures by referring Naselli's figure. Please check our new Figs 2 and 3.

There was a little misunderstanding by us. We exactly know the total number of insects used in our Y-tube test. There were many replications for each insect (N. tenuis - 50 times replicated, 5 replications per each wavelength x 10 test wavelengths; B. tabaci - 25 times replicated; 5 replications per each wavelength x 5 test wavelengths). Thus, we thought that it was hard to express all initial numbers respectively. Now, we understood your intention. So, we expressed the initial number range for each wavelength treatment instead of all numbers in our new Figs 2 and 3. Please check our new Figs 2 and 3 and L245 and 251 in the revised manuscript.

2. Reviewer comment: Change “Exp. #1” to “Experiment 1” throughout all the manuscript, experiments (2, 3, and 4), including the subheadings.

Our response: Yes. Please check L97, 101, 115, 144, 160, 205, 207, 209, 210, 217, 220, 226, 242, 248, 253, 255, 267, 271, and 278 in the revised manuscript.

3. Reviewer comment: Change “branches” by “arms” about the y-tube.

Our response: Yes. Please check L118-120, 123, 126, and 128 in the revised manuscript.

4. Reviewer comment: Why are the ±SE of the abiotic conditions (Temperature and relative humidity) too high?

Our response: That figure was deleted by removing the greenhouse part in the manuscript.

---

## [Editor Report · Decision Letter 2]

23 Dec 2020

UV-LED Lights Enhance the Establishment and Biological Control Efficacy of Nesidiocoris tenuis (Reuter) (Hemiptera: Miridae)

PONE-D-20-30203R2

Dear Dr. Lee,

We’re pleased to inform you that your manuscript has been judged scientifically suitable for publication and will be formally accepted for publication once it meets all outstanding technical requirements.

Kind regards,

Antonio Biondi, Ph.D.

Academic Editor

PLOS ONE
---

## [Editor Report · Acceptance letter]

29 Dec 2020

PONE-D-20-30203R2 

UV-LED Lights Enhance the Establishment and Biological Control Efficacy of *Nesidiocoris tenuis* (Reuter) (Hemiptera: Miridae) 

Dear Dr. Lee:

I'm pleased to inform you that your manuscript has been deemed suitable for publication in PLOS ONE. Congratulations! Your manuscript is now with our production department. 

Kind regards, 

on behalf of

Dr. Antonio Biondi 

Academic Editor

PLOS ONE